# First-order visual interneurons distribute distinct contrast and luminance information across ON and OFF pathways to achieve stable behavior

**Madhura D Ketkar[1,2†], Burak Gür[1,2†], Sebastian Molina-Obando[1,2†], Maria Ioannidou[1], Carlotta Martelli[1], Marion Silies[1]\***

[1]Institute of Developmental Biology and Neurobiology, Johannes-Gutenberg University Mainz, Mainz, Germany; [2]Göttingen Graduate School for Neurosciences, Biophysics, and Molecular Biosciences (GGNB) and International Max Planck Research School (IMPRS) for Neurosciences at the University of Göttingen, Göttingen, Germany

**\*For correspondence:**
msilies@uni-mainz.de

[†]These authors contributed equally to this work

**Competing interest:** The authors declare that no competing interests exist.

**Abstract** The accurate processing of contrast is the basis for all visually guided behaviors. Visual scenes with rapidly changing illumination challenge contrast computation because photoreceptor adaptation is not fast enough to compensate for such changes. Yet, human perception of contrast is stable even when the visual environment is quickly changing, suggesting rapid post receptor luminance gain control. Similarly, in the fruit fly *Drosophila*, such gain control leads to luminance invariant behavior for moving OFF stimuli. Here, we show that behavioral responses to moving ON stimuli also utilize a luminance gain, and that ON-motion guided behavior depends on inputs from three first-order interneurons L1, L2, and L3. Each of these neurons encodes contrast and luminance differently and distributes information asymmetrically across both ON and OFF contrast-selective pathways. Behavioral responses to both ON and OFF stimuli rely on a luminance-based correction provided by L1 and L3, wherein L1 supports contrast computation linearly, and L3 non-linearly amplifies dim stimuli. Therefore, L1, L2, and L3 are not specific inputs to ON and OFF pathways but the lamina serves as a separate processing layer that distributes distinct luminance and contrast information across ON and OFF pathways to support behavior in varying conditions.

## Editor's evaluation

This paper combines silencing and rescue experiments with measurements of cellular responses and behavior to investigate how three early visual neurons in the fly eye encode both scene luminance and scene contrast. It reveals that these neurons carry different information about scene luminance and contrast that gets distributed to ON and OFF selective pathways that guide behavior.

## Introduction

Across species, contrast information forms the basis of visual computations. Contrast is the relative change in luminance, which can be computed across space or across time. For our perception to be stable, our eyes must compute contrast relative to the mean illumination of a scene. In natural environments, illumination changes by several orders of magnitude not only from dawn to dusk, but also at much faster timescales as our eyes saccade across a scene or we quickly move from sun to shade (*Frazor and Geisler, 2006*; *Mante et al., 2005*; *Rieke and Rudd, 2009*). Thus, the computation

of contrast needs to be invariant to rapid changes in luminance, such that visual perception of a given contrast remains constant. Invariant responses to contrast are accomplished by human perception, even when background luminance quickly changes (*Burkhardt et al., 1984*). At the circuit level, neuronal responses in the cat lateral geniculate nucleus (LGN) display luminance-invariant responses at rapid time scales (*Burkhardt et al., 1984*; *Mante et al., 2005*). Thus, robust contrast computation at rapid time scales appears to be a wide-spread phenomenon across visual systems. However, contrast encoding in photoreceptors is not luminance invariant when the stimulus changes more rapidly than photoreceptor adaptation (*Laughlin and Hardie, 1978*; *Normann and Werblin, 1974*), arguing for a common post-receptor corrective mechanism.

In most visual systems, information is split into two separate ON and OFF pathways, that process contrast increments (ON) or contrast decrements (OFF), respectively (*Behnia et al., 2014*; *Franceschini et al., 1989*; *Silies et al., 2014*; *Yang and Clandinin, 2018*). The visual OFF pathway in fruit flies drives luminance-invariant behavior (*Ketkar et al., 2020*). In the OFF pathway, luminance information itself is maintained postsynaptic to photoreceptors, and is crucial for the accurate estimation of temporal contrast, resulting in luminance-invariant behavior. Luminance serves as a corrective signal, leading to a luminance gain that adjusts temporal contrast computation when background luminance quickly changes (*Ketkar et al., 2020*). The requirement of such a corrective signal can be theoretically expected regardless of ON and OFF contrast polarities, since the adaptational constraints in dynamic environments challenge both contrast polarities. However, the ON and OFF pathways are not mere sign-inverted versions of each other since they face different environmental challenges (*Clark et al., 2014*; *Ruderman and Bialek, 1994*) and have evolved several structural and physiological asymmetries (*Chichilnisky and Kalmar, 2002*; *Jin et al., 2011*; *Leonhardt et al., 2016*; *Ratliff et al., 2010*). It is thus not clear if this luminance invariance is a general feature of both ON and OFF pathways, and how luminance and contrast information are distributed across visual pathways to establish luminance invariance.

The *Drosophila* visual system is composed of a columnar arrangement of 800 repeating units, carrying the same set of columnar neurons, together forming a retinotopic map. Different columnar neuronal cell types were assigned to distinct ON or OFF pathways based on physiological properties (*Molina-Obando et al., 2019*; *Serbe et al., 2016*; *Shinomiya et al., 2019*; *Silies et al., 2013*; *Strother et al., 2017*), anatomical connectivity (*Shinomiya et al., 2014*; *Takemura et al., 2015*; *Takemura et al., 2013*; *Takemura et al., 2017*), and behavioral function (*Clark et al., 2011*; *Silies et al., 2013*). ON and OFF contrast selectivity first arises two synapses downstream of photoreceptors, in medulla neurons (*Fischbach and Dittrich, 1989*; *Serbe et al., 2016*; *Silies et al., 2013*; *Strother et al., 2017*; *Yang et al., 2016*). In each visual column, these medulla neurons receive photoreceptor information through the lamina neurons L1-L3, together referred to as large monopolar cells (LMCs). LMCs project to specific medulla layers (*Meinertzhagen and O'Neil, 1991*; *Strother et al., 2014*). Although all LMCs show the same response polarity and hyperpolarize to light onset and depolarize to light offset, L1 projects to layers where it mostly connects to ON-selective medulla neurons. Similarly, L2 and L3 project to layers where OFF-selective medulla neurons get most of their inputs (*Shinomiya et al., 2014*; *Takemura et al., 2015*; *Takemura et al., 2013*). L1 is thus thought to be the sole major input of the ON pathway, whereas L2 and L3 are considered the two major inputs of the OFF pathway (*Figure 1A*; *Clark et al., 2011*; *Joesch et al., 2010*; *Shinomiya et al., 2019*). Among these, L2 is contrast sensitive, but cannot support luminance invariance alone when photoreceptor adaptation is insufficient. Instead, the stable computation of contrast at changing background luminance in OFF-motion guided behavior (OFF behavior) is ensured by a corrective signal from luminance-sensitive L3 neurons (*Ketkar et al., 2020*). It is not known whether ON-motion-driven behavior (ON behavior) also requires a post-receptor luminance gain and whether L1 can provide it along with its contrast signal (*Figure 1B*).

Contrast and luminance are encoded by the transient and sustained response components in both vertebrates and invertebrate photoreceptors, respectively (*Laughlin and Hardie, 1978*; *Normann and Perlman, 1979*; *Normann and Werblin, 1974*; *Shapley and Enroth-Cugell, 1984*), which are captured differentially by their downstream neurons. In the vertebrate retina, many different types of first order interneurons, bipolar cells, exist. Although they are generally thought to capture the contrast component of the photoreceptor response, luminance information has been shown to be preserved in visual circuitry postsynaptic to photoreceptors (*Awatramani and Slaughter, 2000*; *Ichinose and Hellmer,*

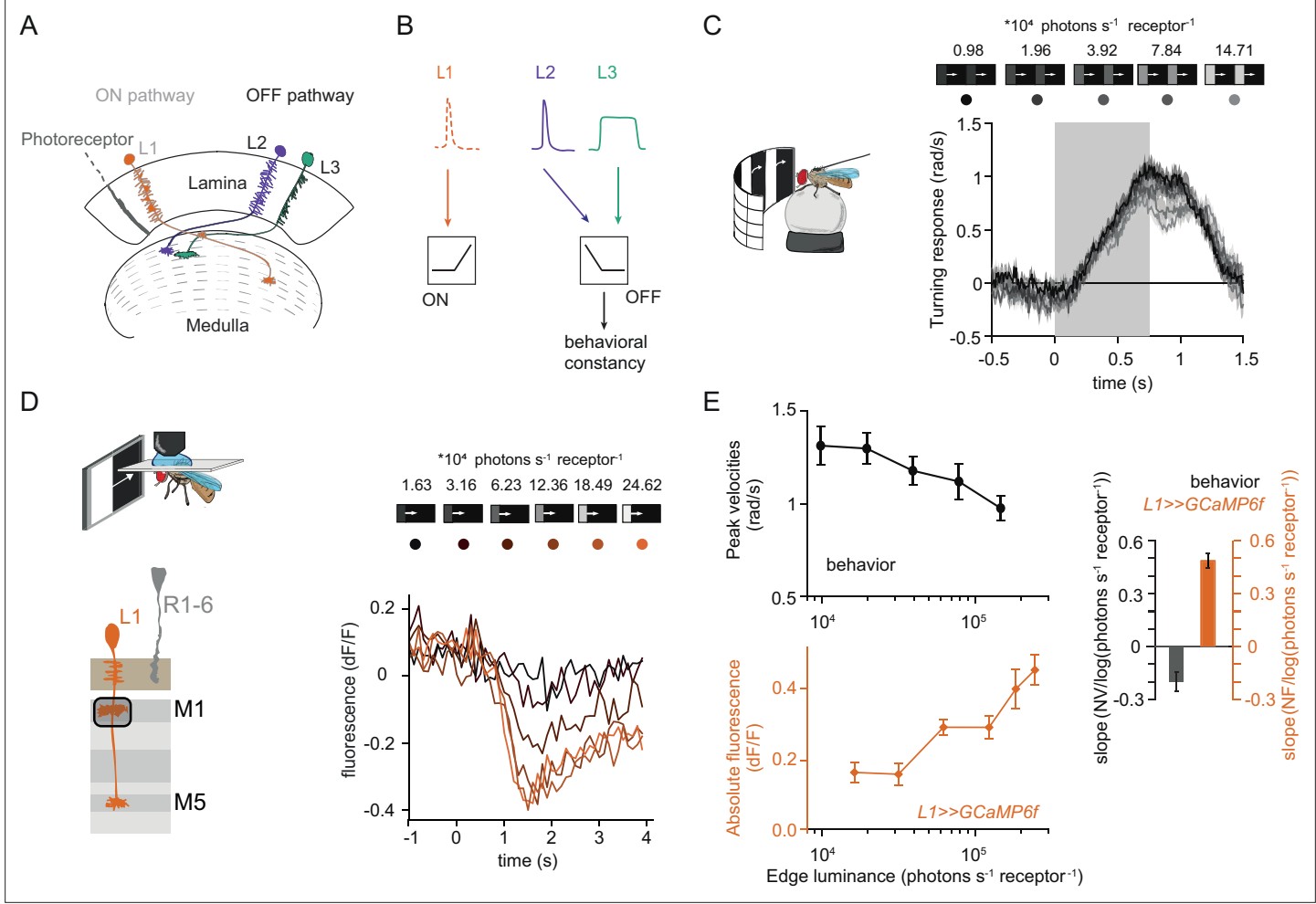

**Figure 1.** Fly behavioral responses to ON contrast do not co-vary with L1 responses. (**A**) Schematic of lamina neurons projecting from the lamina to the medulla. L1 is considered the main input to the ON-pathway, whereas L2 and L3 are thought to provide input to the OFF pathway. (**B**) Transient L2 and sustained L3 neurons provide contrast and luminance information, respectively, to the OFF pathway to guide contrast-constant behavior (*Ketkar et al., 2020*). L1 is thought to have physiological properties very similar to L2 (*Clark et al., 2011*) and provides contrast information to the ON selective pathway. (**C**) Turning response to multiple moving ON edges, moving at 160°/s, displayed on an LED arena that surrounds a fly walking on an air-cushioned ball. The edge luminance takes five different values, and the background is dark (~0 luminance), all resulting in 100% contrast. Turning responses are color-coded according to the edge luminance. The gray box indicates motion duration. n = 10 flies. (**D**) In vivo calcium signals of L1 axon terminal in medulla layer M1 in response to moving ON edges of six different luminances. Calcium responses of single L1 axon terminal are shown. (**E**) Top: peak turning velocities calculated from (**C**), bottom: absolute step responses of L1. Sample size for L1: n = 6 (15) flies(cells). Right: slope quantification of luminance dependency for normalized behavior and L1 fluorescence signals. NV = normalized peak velocity, NF = normalized fluorescent signal. Traces and plots in C and E show mean ± SEM.

*2016*; *Ichinose and Lukasiewicz, 2007*; *Odermatt et al., 2012*; *Oesch and Diamond, 2011*). As suggested by their sustained response component, different degrees of luminance-sensitivity exist across bipolar cell types (*Baden et al., 2016*; *Euler et al., 2014*). Furthermore, ON and OFF contrast selectivity emerges at the bipolar cell layer, where ON selectivity emerges through glutamatergic inhibition (*Masu et al., 1995*). These ON and OFF bipolar cells also split anatomically, as they innervate different layers (*Euler et al., 2014*).

Together, many parallels exist between the *Drosophila* visual system and the vertebrate retina, including the response properties of photoreceptors, the layered organization and the existence of ON and OFF pathways (*Clark and Demb, 2016*; *Mauss et al., 2017*). However, in contrast to the vertebrate retina, fewer first-order interneuron types distribute contrast and luminance information, and contrast selectivity itself only occurs one synapse further downstream, where neurons postsynaptic to lamina neurons are either ON or OFF selective. Comparing the vertebrate retina with the

insect visual system, it is unclear how just three first-order interneurons distribute their different physiological properties across visual pathways.

Here, we show that luminance and contrast information are distributed to and are of behavioral relevance for both ON and OFF pathways. In vivo calcium imaging experiments reveal that each first-order interneuron is unique in its contrast and luminance encoding properties. Although L2 is purely contrast sensitive, L1 encodes both contrast and luminance in distinct response components. L1 linearly scales with luminance, whereas the luminance-sensitive L3 non-linearly amplifies dim light. Behavioral experiments further show that these differential luminance- and contrast- encoding properties translate into distinct behavioral roles. In the ON pathway, L1 and L3 both provide a luminance gain that scales behavioral responses to contrast. Furthermore, L2, known as the OFF-pathway contrast input, provides contrast information to the ON-pathway, in addition to L1. Surprisingly, both L1 and L3 neurons are necessary and sufficient for OFF behavior. These findings indicate that L1, L2, and L3 do not constitute ON- or OFF-specific inputs. Instead, the three first-order interneurons encode luminance and contrast differentially and contribute to computations in both ON and OFF pathways. Together, our data reveal how luminance and contrast information are distributed to both ON and OFF pathways to achieve stable visual behavior.

## Results

### L1 responses to contrast do not explain ON behavior

Luminance-invariant visual responses have been observed in multiple species (*Burkhardt et al., 1984*; *Mante et al., 2005*), highlighting their relevance. In *Drosophila*, luminance-invariant behavior has been shown in response to moving OFF edges, where a dedicated luminance-sensitive pathway scales contrast-sensitive inputs to achieve luminance invariance in behavior (*Ketkar et al., 2020*). The ON pathway is thought to have just one prominent input, L1. We thus asked if luminance-invariant behavior is achieved in the ON pathway and if this can be accounted for by the contrast-sensitive input L1. For this purpose, we first compared turning behavior of walking flies with the responses of L1. Behavioral responses were measured in a fly-on-a-ball assay. Flies were shown moving ON edges of different luminance but the same 100% Michelson contrast ($C_M = (I_{edge}-I_{background}) / (I_{edge} +I_{background})$, where C stands for contrast and I stands for luminance). Consecutive motion epochs were separated by a dark interstimulus interval. Fly turning responses were similar across luminances, with low-luminance edges eliciting slightly larger turning responses than brighter edges (*Figure 1C*).

We wondered if the sole known ON-pathway input L1 can directly drive this behavior. To test this, we examined the contrast responses of L1 to moving ON edges with comparable parameters and overlapping luminance values as those used in the behavioral assay (*Figure 1D*). We recorded L1 in vivo calcium responses to visual stimuli from its axon terminals expressing GCaMP6f using two-photon microscopy. As described previously, L1 responded negatively to contrast increments, in line with the inverted response polarity of lamina neurons (*Figure 1D*; *Clark et al., 2011*; *Laughlin and Hardie, 1978*; *Yang et al., 2016*). The absolute response amplitude of the L1 calcium signals scaled with luminance, showing smaller response in low as compared to high luminances, and did not co-vary with the behavioral response (*Figure 1E*). To extract the luminance dependency of the response, we performed linear regression across calcium signals at different luminances and quantified the slope. L1 signals and behavioral responses had opposite luminance dependencies (*Figure 1E*). Thus, the observed behavior cannot be explained solely by contrast inputs from L1, suggesting that the ON pathway additionally gets a luminance-sensitive input.

### L1 neuronal responses carry a luminance-sensitive component

To explore the source of luminance information in first-order interneurons, we measured calcium signals in L1, L2, and L3. Flies were shown a staircase stimulus with luminance going sequentially up and down. L1 and L2 showed transient negative responses when luminance stepped up, and transient-positive responses when luminance stepped down (*Figure 2A*), consistent with the contrast sensitivity described for L1 and L2 (*Clark et al., 2011*; *Silies et al., 2013*). L2 did not show any sustained component. L3 showed sustained responses to OFF steps and was non-linearly tuned to stimulus luminance, responding strongly to the darkest stimulus. Intriguingly, L1 showed a transient component followed by a sustained component, suggesting that it encodes luminance in addition to contrast (*Figure 2A*).

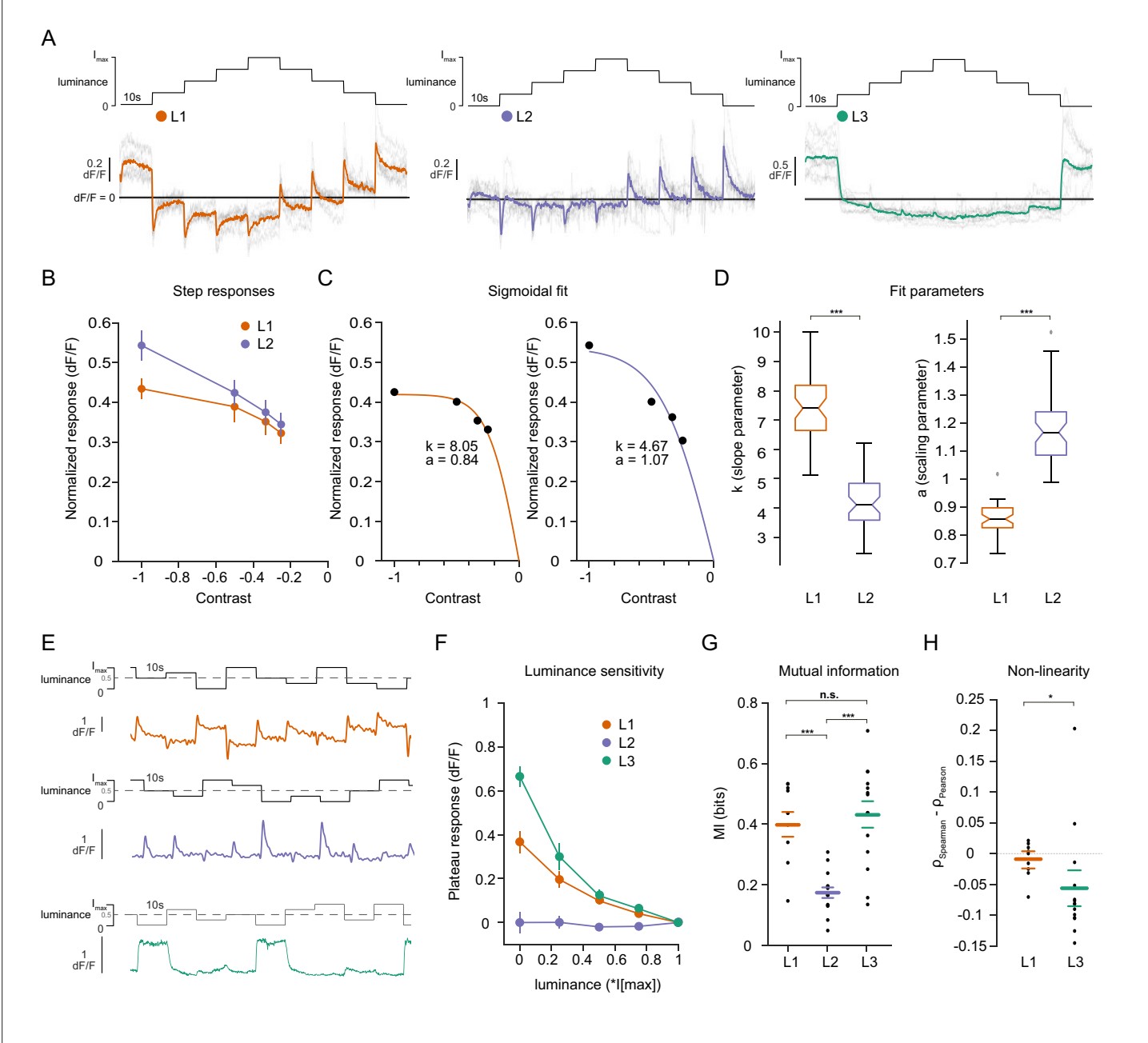

**Figure 2.** Lamina neuron types L1-L3 are differently sensitive to contrast and luminance. (**A**) Schematic of the 'staircase' stimulus, $I_{max}$ = 2.17 × 10⁵ photons s⁻¹ photoreceptor⁻¹. Luminance sequentially steps up through five values and then sequentially steps down. Shown below are the calcium responses of L1 (orange), L2 (purple), and L3 (green) axon terminals. Colored traces show the mean response, grey traces show individual fly means. (**B**) Normalized step responses of L1 and L2 neurons to OFF contrasts of the staircase stimulus. (**C**) Individual bootstrapping examples of sigmoidal fits where k is the slope parameter and a is the scaling parameter. (**D**) Comparison of fit parameters between L1 and L2, Student t-test, ***p < 0.001 (**E**) Example calcium traces of single L1, L2, and L3 axon terminals to a stimulus comprising 10 s full-field flashes varying randomly between five different luminances. (**F**) Plateau responses of the three neuron types, quantified from the responses to the stimulus in (**E**). (**G**) Mutual information between luminance and calcium signal, ***p < 0.001, one-way ANOVA followed by multiple comparison test corrected with Bonferroni. (**H**) Non-linearity quantification of luminance-dependent signals of L1 and L3 in (**C**), *p < 0.05, tested by a wilcoxon rank sum test. Sample sizes for (**A and B**) L1: n = 8 (54), L2: n = 11(48), L3: n = 12(103), for (**D**) we used 50 times bootstrapping from the dataset in (**A**), and for (**E–H**) L1: n = 9 (71), L2: n = 14(74), L3: n = 14(88) flies(cells). Calcium traces show mean and quantification plots (**B, F, G, H**) show mean ± SEM. Boxplots in (**D**) show median, 25% and 75% percentiles and whiskers extend to the most extreme data points.

The online version of this article includes the following figure supplement(s) for figure 2:

**Figure supplement 1.** L1 has contrast and luminance-sensitive components.

The sustained components of L1 responses were negatively correlated with luminance, such that the baseline calcium signal at each step sequentially increased with decreasing stimulus luminance. We next used a stimulus that allows to systematically tease apart the contrast and luminance sensitivities of neurons (*Ketkar et al., 2020*; *Oesch and Diamond, 2011*, *Figure 2—figure supplement 1A*). Here, an adapting bright background was followed by two sequential OFF steps. The first step (A step) varied in its contrast and luminance values, whereas the second step (B step) always comprised 25% Weber contrast ($C_W = (I_B − I_A)/I_A$) but took on different luminance values. The transient peak responses of L1 neurons correlated positively with the A step contrast and were indistinguishable for the B steps. This suggests that L1, like L2, encodes contrast in its peak response (*Figure 2—figure supplement 1A, B*, tested with one-way ANOVA), (*Ketkar et al., 2020*). The sustained component of L1 responses negatively correlated with the luminance values of both A and B steps, indicating luminance encoding (*Figure 2—figure supplement 1A,C*). Together, L1 neurons encode contrast in their peak responses and luminance in their sustained responses.

We next explicitly compared contrast and luminance encoding between the input neurons. To look at the contrast encoding properties, we analyzed the step responses of L1 and L2 neurons to different OFF contrasts in the staircase stimulus (*Figure 2A and B*). Although L1 and L2 responded similarly to low-contrast stimuli, L2 responses tended to be higher for high contrasts (*Figure 2B*). We fitted sigmoidal contrast response functions (*Figure 2C*) and quantified contrast encoding properties in two ways: First, we used the slope parameter (k) of the sigmoid, indicating steepness of the contrast function, to analyze the encoding of different contrasts (see Materials and methods, *Figure 2C and D*). Second, the scaling parameter (a) shows how much of the available response range is used to encode contrast (*Figure 2C and D*). L1 neuron contrast functions had steeper slopes and thus, L1 reached saturation at lower contrasts than L2, suggesting that L1 neurons encode contrasts with less resolution than L2 neurons do, especially in the high-contrast regimes. Furthermore, analysis of the scaling parameter revealed that L1 neurons dedicated less response range to encode contrast than L2 neurons (*Figure 2C and D*). These results show that L1 and L2 neurons have different contrast encoding properties, arguing that they might fulfill different roles in the circuitry. To look at luminance encoding properties, we measured responses to randomized luminance and calculated the mutual information between stimulus and the sustained response component (*Figure 2E–G*). As for the staircase stimulus, L2 transient responses returned to baseline within the 10 s of the stimulus presentation, whereas both L1 and L3 displayed sustained components that varied with luminance (*Figure 2E and F*). Sustained response components in L1 and L3 carried similar mutual information with luminance, and both were higher than L2 (*Figure 2G*). The luminance-sensitive response components of L1 and L3 scaled differently with luminance. We quantified non-linearity using the difference of Pearson's linear and Spearman's correlation between response and luminance. This value will approach zero if the relationship is linear and increase or decrease if non-linear, depending on the sign of correlation between luminance and response. L1 responses were more linear with respect to luminance than L3 responses, which selectively amplified low luminance (*Figure 2H*). Thus, the two luminance-sensitive neurons carry different types of luminance information.

## L1 is not required but sufficient for ON behavior across luminances

Since the canonical ON-pathway input L1 is also found to carry luminance information, we hypothesized that it plays a role in mediating the observed behavior. To test this, we silenced L1 outputs while measuring ON behavior using Shibire[ts] (*Kitamoto, 2001*). L1 silencing had little effect on responses to 100% contrast at varying luminance, suggesting the existence of other ON-pathway inputs (*Figure 3A and B*). This was initially surprising, considering that previous behavioral studies identified L1 as the major input to the ON pathway (*Clark et al., 2011*; *Silies et al., 2013*). L1 silenced flies also turned normally to ON edges of fixed luminance, ruling out the possibility that changing luminance underlies this inconsistency (*Figure 3—figure supplement 1A, B*). However, L1 silencing severely reduced turning responses when a bright instead of a dark inter-stimulus interval was used, explaining the discrepancy between this and previous studies (*Figure 3—figure supplement 1C, D*). Thus, L1 is indeed a major but not the sole input to the ON pathway.

To explicitly test if and how L1 silencing changed the luminance dependence of behavioral responses, we quantified the slope of peak turning velocities across different background luminances (*Figure 3C*). The slopes were slightly negative for both the control and L1-silenced conditions, and did

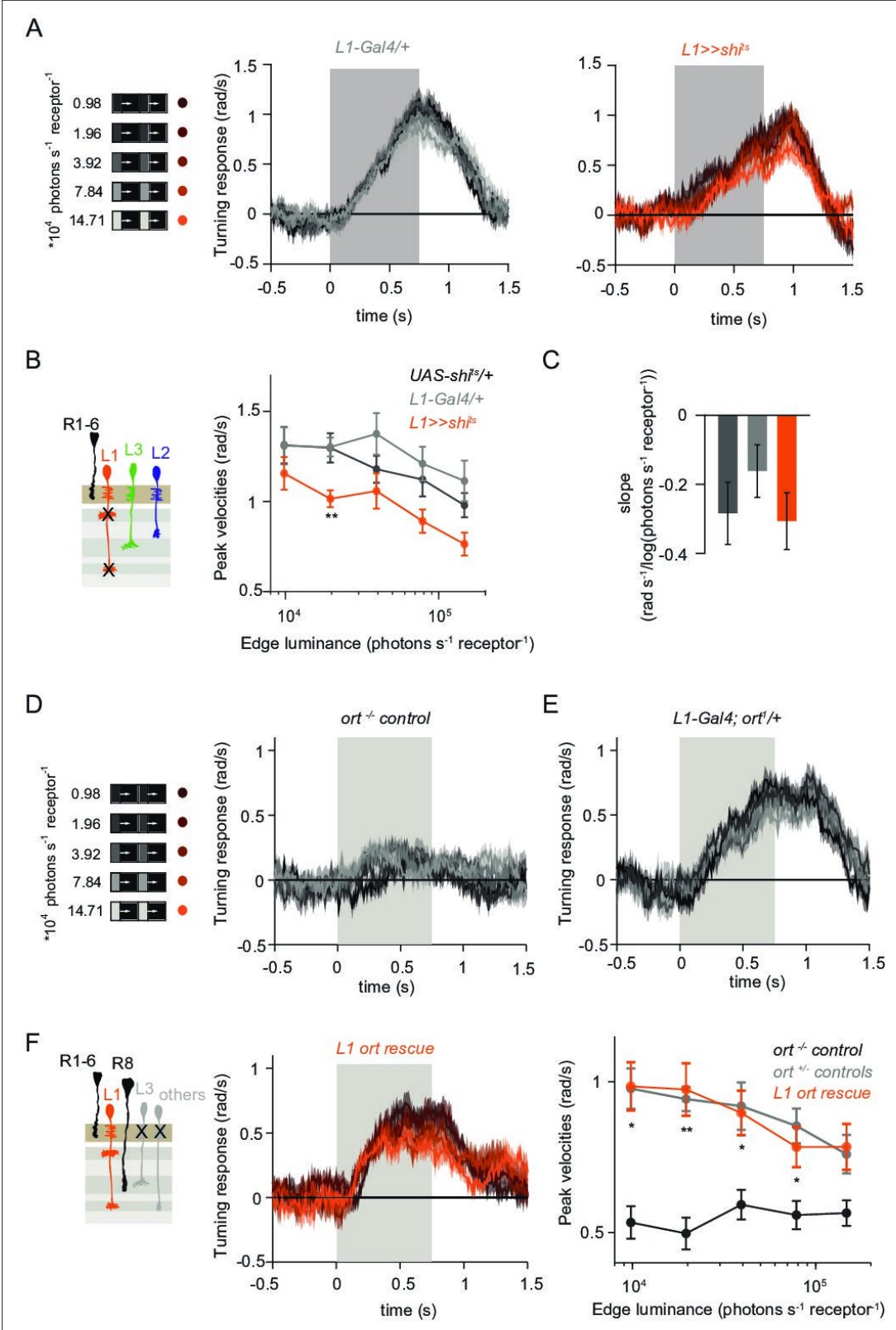

**Figure 3.** L1 is not required but sufficient for ON behavior across luminance. (**A**) Turning responses of L1-silenced flies (orange) and their specific Gal4 control (gray) to moving 100% contrast ON edges at five different luminances. (**B**) Peak velocities quantified for each of the five edges during the motion period, also including the control *UAS-shi^{ts}/+*, \*\*p < 0.01, two-tailed Student's t-tests against both controls, with Bonferroni-Holm correction. (**C**) Relationship of the peak velocities with luminance, quantified as slopes of the linear fits to the data in (**B**). Sample sizes are n = 10 flies for each

*Figure 3 continued on next page*

*Figure 3 continued*

genotype. (**D–E**) Schematic of the stimulus (same as in A) and turning responses of the *ort* null mutant (*ort*[-/-] controls, **D**) and heterozygous *ort* controls (*ort*[+/-]±, **E**). (**F**) Schematic of the L1 ort rescue genotype and turning responses of L1 ort rescue flies (left). Peak turning velocities of L1 ort rescue flies and the respective controls (right); *p < 0.05, **p < 0.01, two-tailed Student's t-tests against both controls, with Bonferroni-Holm correction. The gray box region in (**A,D,E,F**) indicates motion duration. Traces and plots show mean ± SEM.

The online version of this article includes the following figure supplement(s) for figure 3:

**Figure supplement 1.** L1 is required for ON behavior across a range of contrasts.

not differ significantly between conditions, suggesting another luminance input masking the L1 contribution. To test this possibility, we asked if L1 is sufficient to contribute to ON behavior in dynamically changing luminance conditions. We measured behavioral responses after functionally isolating L1 from other circuitry downstream of photoreceptors. To achieve this, we selectively rescued expression of the histamine-gated chloride channel Ort (Ora transientless) in *ort*-mutant flies, which otherwise lack communication between photoreceptors and their postsynaptic neurons. Behavioral responses of *ort* mutant control flies were absent, indicating that ON-motion behavior fully depends on Ort (*Figure 3D*). Heterozygous *ort* controls turned with the moving 100% contrast ON edges at all luminances (*Figure 3E*). Flies in which *ort* expression was rescued in L1 responded to ON motion at all luminances, and indistinguishable from controls (*Figure 3F*), showing that L1 can mediate normal turning behavior to ON edges at all luminances. This data confirms L1's general importance in the ON pathway.

## L1 and L3 together provide luminance signals required for ON behavior

Our data suggest the existence of a second luminance input to the ON pathway. In the OFF pathway, the luminance-sensitive L3 neuron provides the necessary luminance-based correction to achieve contrast constancy (*Ketkar et al., 2020*). Connectomics data suggest that L3 could provide input to the ON pathway as well, as it makes direct synaptic contact with the major ON-pathway medulla neurons Mi1 and Mi9 (*Takemura et al., 2013*). To test the hypothesis that L3 also provides a luminance signal to the ON pathway, we measured behavioral responses to a set of 100% contrast ON edges at five different luminances while silencing L3 synaptic outputs (*Figure 4A–C*). Interestingly, unlike controls, L3-silenced flies responded stronger to all ON edges, revealing a potential, unexplored role of L3 in inhibiting behavioral responses to certain stimuli. However, the responses of L3-silenced flies were still similar across luminances (*Figure 4A and B*). Unlike controls, L3 silenced flies did not show a slight increase in turning amplitude at lower edge luminance, also reflected in the differences in their slopes (*Figure 4C*), suggesting that L3 inputs to the ON pathway also contribute to behavior in a luminance-dependent manner. To further explore if L3 indeed serves as an ON-pathway input, we next asked if L3 is sufficient for ON behavior and functionally isolated L3 from other circuitry. L3 ort rescue flies turned to ON edges at all luminances tested (*Figure 4D*) and significantly rescued turning behavior at low luminances compared to *ort* mutant flies (*Figure 4E*), showing that L3 is sufficient for ON behavior at low luminances. This further reflects L3's nonlinear preference for dim light seen at the physiological level (*Ketkar et al., 2020*, *Figure 2F*).

We found that L3 is a second luminance input to the ON-pathway. To ask if L3, together with L1, provides a luminance gain to scale ON behavior, we simultaneously silenced the outputs of both L1 and L3 while measuring ON behavior across luminance. Flies still turned to the moving ON edges. However, unlike control responses which slightly deviated from luminance invariance by showing a negative correlation with luminance, turning responses of flies lacking both L1 and L3 functional outputs were positively correlated with luminance (*Figure 4F–H*). Intriguingly, behavioral responses of flies lacking both lamina neurons carrying luminance information underestimated dim stimuli (*Figure 4G*), and qualitatively recapitulated the LMC contrast-sensitive responses (*Ketkar et al., 2020*). Thus, L1 and L3 can together account for the luminance information available to the ON pathway. To analyze the extent of the individual contributions of L1 and L3, we compared L1 and L3 ort rescues by computing rescue efficiency, defined as the fraction of the difference between positive and negative control behaviors. Whereas L1 fully rescued turning behavior to ON edges at all luminances, L3 significantly rescued turning behavior selectively at low luminances (*Figure 4I*). Taken together, L3 is a functional input to the ON pathway, to which L1 and L3 both provide distinct types

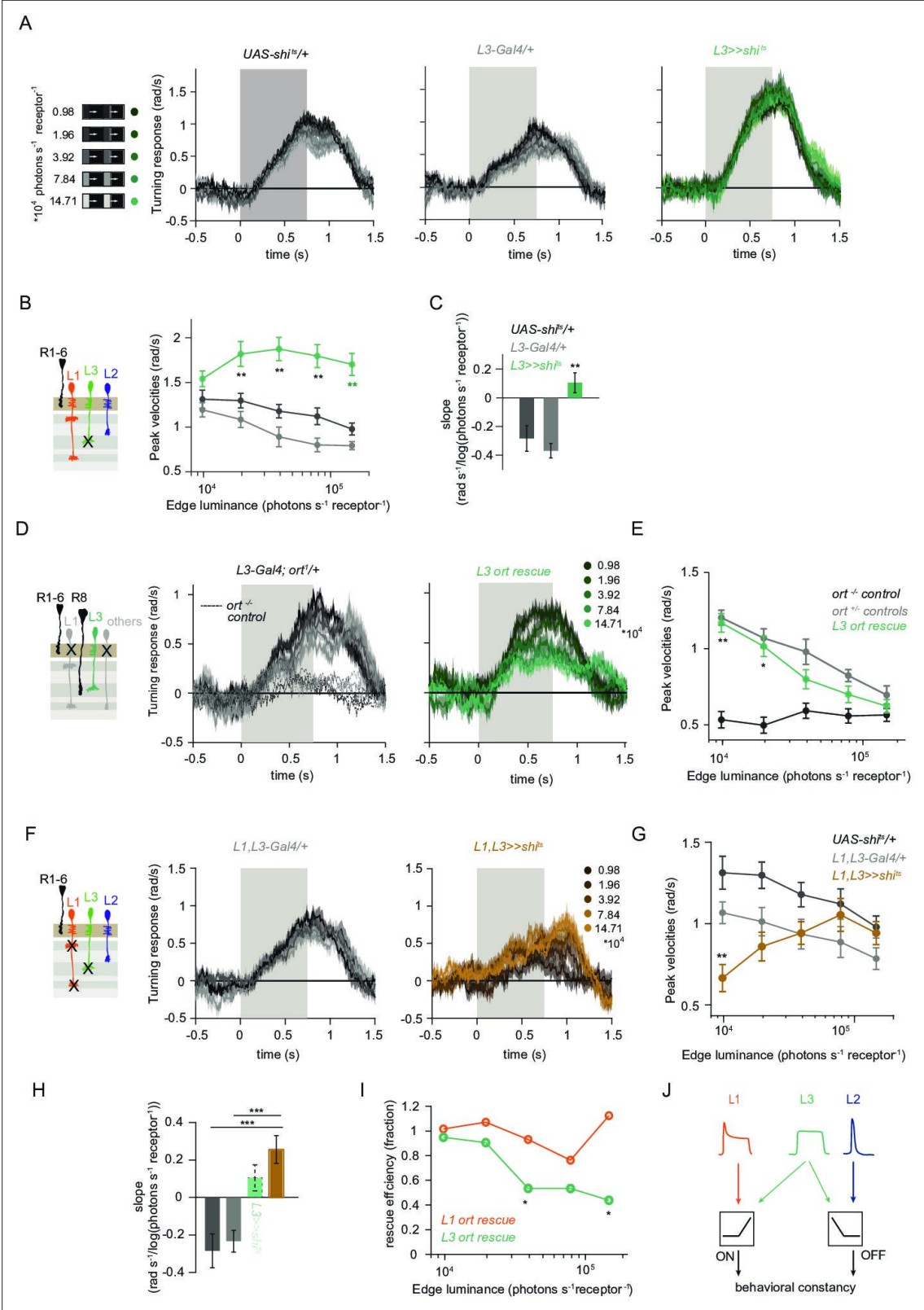

**Figure 4.** L1 and L3 together provide luminance signals required for ON behavior. (**A**) Turning velocities of the controls (gray) and L3-silenced flies (green) in response to five moving ON edges of 100% contrast. The gray box region indicates motion duration. (**B**) Peak turning velocities for five ON edges quantified during the motion period, **p < 0.01, two-tailed Student's t-tests against both controls, with Bonferroni-Holm correction. (**C**) Relationship of the peak velocities with luminance, quantified as slopes of the linear fits to the data in (**B**). Fitting was done for individual flies. Sample

*Figure 4 continued on next page*

*Figure 4 continued*

sizes are n = 10 (*UAS-shi^{ts}/+,L3 >>shi^{ts}*) and n = 8 (*L3^{0595}-Gal4/+*). **p < 0.01, two-tailed Student's t-tests against both controls, with Bonferroni-Holm correction. (**D**) Schematic of the L3 *ort* rescue genotype and turning responses of the heterozygous control (gray) and rescue (green) flies. (**E**) Peak turning velocities, *p < 0.05, **p < 0.01, two-tailed Student's t-tests against both controls, with Bonferroni-Holm correction. (**F**) Turning responses of flies where L1 and L3 were silenced together (golden brown) and their specific Gal4 control (gray), color-coded according to ON edge luminance. The same five moving ON edges of 100% contrast as in *Figure 1C* were shown. Responses of the other control *UAS-shi^{ts}/+* to these stimuli have been included in *Figure 1C*. (**G**) Peak velocities quantified for each of the five edges during the motion period, also including the control *UAS-shi^{ts}/+*, **p < 0.01, two-tailed Student's t-tests against both controls, with Bonferroni-Holm correction (**H**) Relationship of the peak velocities with luminance, quantified as slopes of the linear fits to the data in (**G**). Slopes from the L3-silenced flies (green, dashed) responding to the same stimuli (*Figure 3C*) are included again for comparison. Fitting was done for individual flies. Sample sizes are n = 10 (*UAS-shi^{ts}/ +* and *L1,L3 >>shi^{ts}*) and n = 7 (*L1^{c2025}-Gal4/+;L3^{0595}-Gal4/+*). (**I**) Efficiency of the L1 and L3 behavioral rescue, calculated for each edge luminance as (*rescue - ort^{-/-} control*) / (*ort^{+/-} control - ort^{-/-}* control). ± < 0.05, permutation test with Bonferroni correction, 1,000 permutations over the *L1 ort rescue* and *L3 ort rescue* flies. (**J**) Summary schematic. The ON pathway in addition to the OFF pathway receives a prominent input from L3. Like the OFF pathway, the ON pathway drives contrast constant behavior. Traces and plots show mean ± SEM.

of luminance information (*Figure 4J*). Because flies lacking both neurons still respond to moving ON edges, our data suggest the existence of an unidentified contrast input.

## The contrast-sensitive L2 neuron provides input to the ON-pathway

Besides L1 and L3, the remaining input downstream of photoreceptors is the contrast-sensitive L2 neuron, which provides strong inputs to OFF-pathway neurons (*Takemura et al., 2013*). To explore the possibility of L2 also being an ON-pathway input, we silenced L2 outputs either individually or together with L1. L2-silenced flies showed only slightly reduced turning to all ON edges as compared to controls (*Figure 5A and B*) similarly to silencing L1 alone (*Figure 3A and B*). However, when L1 and L2 were silenced together, fly turning responses were fully disrupted across conditions (*Figure 5C and D*). Moreover, these flies did not turn to moving ON edges of other contrasts either (*Figure 5—figure supplement 1*). This shows that L2, together with L1, is required for ON behavioral responses across different contrasts and luminances. Altogether, L1, L2, and L3 are all ON-pathway inputs.

## L1 is also an OFF-pathway input

Given that three lamina neuron inputs encode visual stimuli differently and that all of them convey information to the ON-motion pathway, we next asked if L1 could also contribute to OFF-pathway function. To test if L1 contributes a luminance gain to the OFF pathway, we silenced L1 neurons while showing moving OFF edges, all of –100% contrast, and moving across five different background luminances. Although the two controls showed overall different response amplitudes, both controls showed luminance-invariant responses (*Figure 6A and B*). Previous work showed that L3 is required to achieve luminance invariance by scaling behavioral responses when background luminance turned dark (*Ketkar et al., 2020*). Similarly, when L1 was silenced, behavioral responses were no longer invariant across luminance, but flies turned less to –100% contrast at low luminance as compared to high luminance (*Figure 6A and B*). Underestimation of the dim OFF edges by L1-silenced flies was not as strong as by L3-silenced flies (*Ketkar et al., 2020*), again highlighting the specialized role of L3 in dim light (*Figure 6B*). These data demonstrate that L1 inputs provide a luminance gain to the OFF pathway. Since L1 carries both contrast and luminance information, it could also be sufficient to drive OFF behavior. To test this, we measured behavioral responses to OFF edges in L1 ort rescue flies. Heterozygous *ort* controls showed turning responses to –100% OFF edges at five different luminances (*Figure 6C*). As described previously (*Ketkar et al., 2020*), *ort* null mutants were not completely blind to this OFF-edge motion stimulus and responded especially at high luminance but very little at low luminances. L1 ort rescue flies responded similarly to positive controls at low luminances, rescuing responses to OFF edges at dim backgrounds (*Figure 6C and D*). Therefore, L1 is even sufficient to guide OFF behavior under the same conditions that were previously described for L3 (*Ketkar et al., 2020*). Taken together, these findings reveal that the lamina neurons L1 and L3 provide behaviorally relevant information to both ON and OFF pathways. In sum, our data uncover L1, L2, and L3 as important inputs for both ON and the OFF pathways, relevant for visually guided behaviors across luminances (*Figure 6E*).

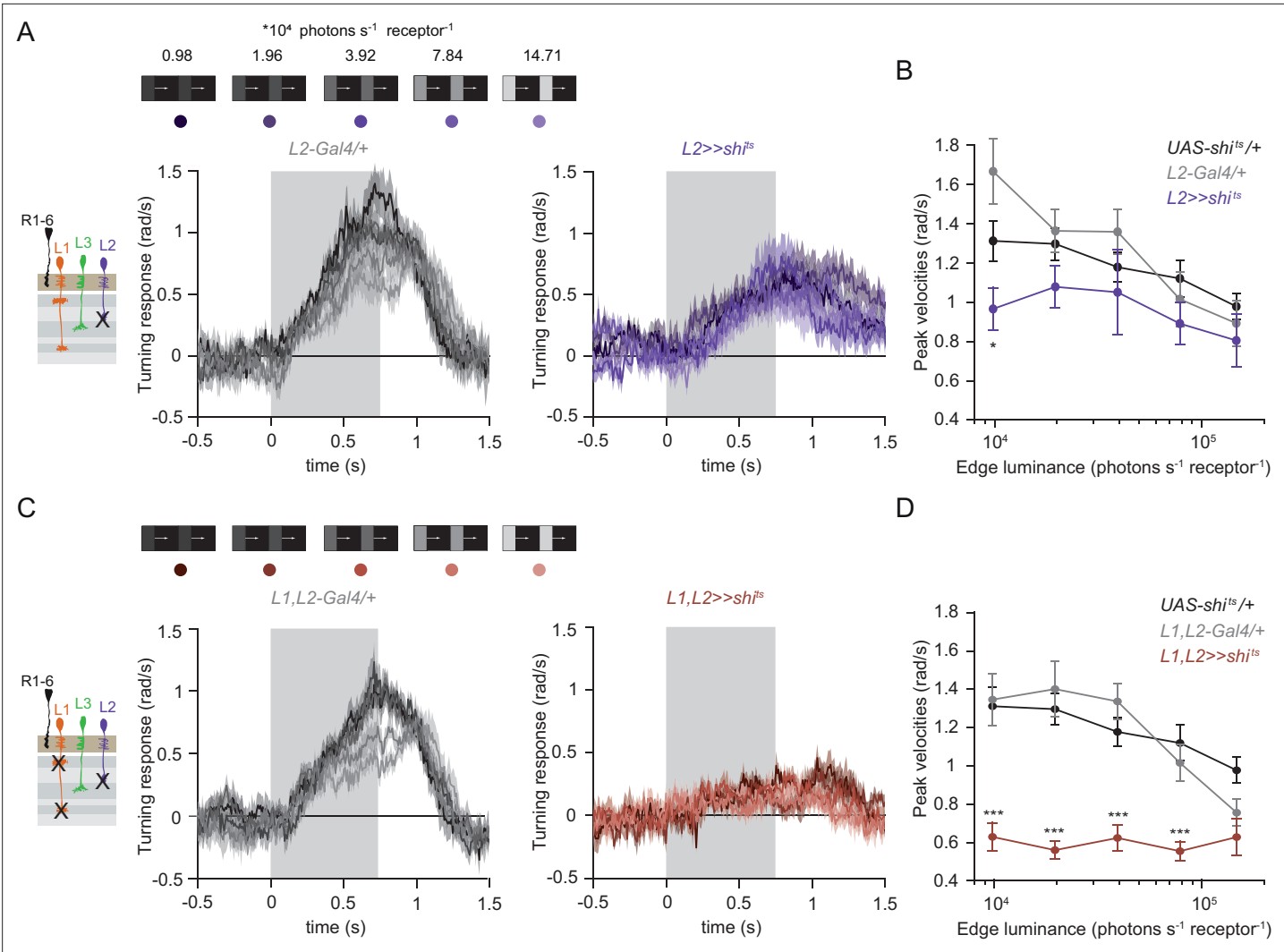

**Figure 5.** The contrast-sensitive L2 provides input to the ON-pathway. (**A**) Turning responses of flies where L2 was silenced (purple) and their specific Gal4 control (gray), color-coded according to 100% contrast ON edge at five different luminances. Sample sizes are n = 9 ($L2^{21Dhh} >> shi^{ts}$) and n = 6 ($L2^{21Dhh}$-Gal4/+). (**B**) Peak velocities quantified for each of the five edges during the motion period, *p < 0.05, two-tailed Student's t tests against both controls, with Bonferroni-Holm correction. (**C**) Turning responses of flies where L1 and L2 were silenced together (brown) and their specific Gal4 control (gray), color-coded according to ON edge luminance. Sample sizes are n = 9 ($L1^{c2025},L2^{21Dhh} >> shi^{ts}$) and n = 8 ($L1^{c2025}$-Gal4/+;$L2^{21Dhh}$-Gal4/+). (**D**) Peak velocities quantified for each of the five edges during the motion period, ***p < 0.001, two-tailed Student's t-tests against both controls, with Bonferroni-Holm correction. Traces and plots show mean ± SEM.

The online version of this article includes the following figure supplement(s) for figure 5:

**Figure supplement 1.** L1 and L2 together are required for ON behavior across a range of contrasts.

## Discussion

The present study establishes that contrast and luminance are basic visual features that interact with both ON and OFF pathways. In both pathways, the interaction between these features enables stable visual behaviors across changing conditions. The lamina neurons L1, L2, and L3 act as the circuit elements segregating both contrast and luminance information. Behavioral experiments show that luminance-sensitive input neurons scale behavioral responses to contrast in both ON and OFF pathways. While L1 and L2 provide distinct contrast inputs, L1 also encodes luminance, together with L3. Whereas L3 activity non-linearly increases with decreasing luminance, L1 shows a linear relationship with luminance. Input from both luminance-sensitive neurons is differently used in ON and OFF pathways. Thus, L1, L2, and L3 are not ON or OFF pathways specific inputs, but they instead distribute

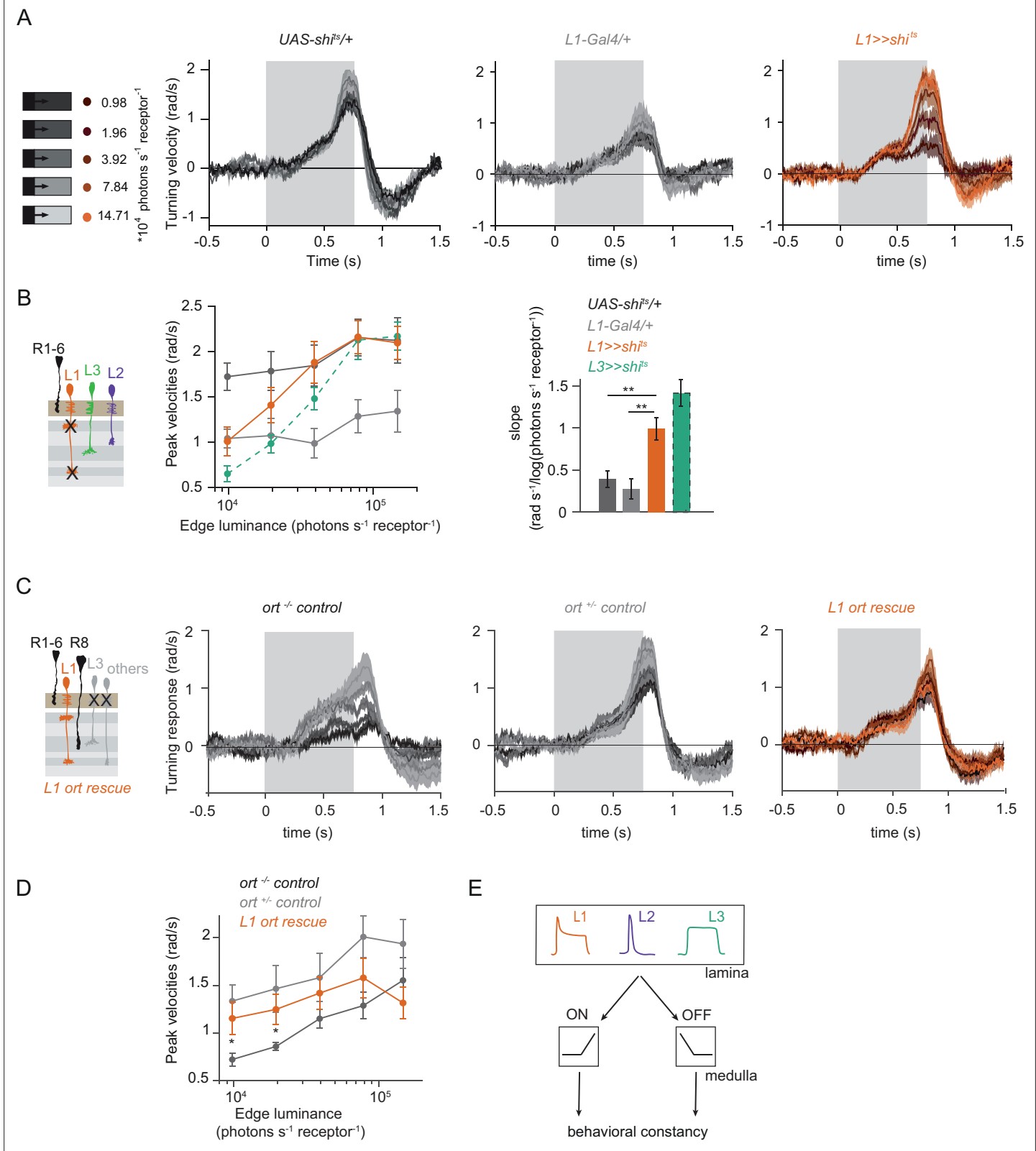

**Figure 6.** L1 function is required and sufficient for OFF behavior. (**A**) Turning responses of L1-silenced flies (orange) and the controls (gray) to five OFF edges moving onto different backgrounds. (**B**) Peak velocities quantified for each of the five edges during the motion period, also including the peak velocities of L3-silenced flies. Shown next to it is the relationship of the peak velocities with luminance, quantified as slopes of the linear fits to the data. **p < 0.01, two-tailed Student's t-tests against both controls, with Bonferroni-Holm correction (not significant against the L3 >>shi$^{ts}$ slopes). UAS-shi$^{ts}$

*Figure 6 continued on next page*

*Figure 6 continued*

data in (**A**) and *ort⁻/⁻* data in (**C**) have been adapted from Figure 1B and 7C in **Ketkar et al., 2020**, *L3 >>shiᵗˢ* data in (**B**) have been re-quantified from data shown in Figure 4B in **Ketkar et al., 2020**. Sample sizes are n = 7 (*L1-Gal4/+*) and n = 10 for other genotypes. (**C**) Schematics of the L1 ort rescue genotypes followed by its turning responses to the moving OFF edges. (**D**) Peak turning velocities of L1 ort rescue flies and the respective controls; *p < 0.05, two-tailed Student's t-tests against both controls, with Bonferroni-Holm correction. Sample sizes are n = 11 flies (*ort⁻/⁻*control) and n = 10 for other genotypes. The gray box region in (**A**) and (**C**) indicates motion duration. (**E**) Summary schematic. Lamina neurons L1-L3 distribute different visual features necessary for both ON and OFF pathways to guide contrast-constant behavior. Traces and plots show mean ± SEM.

the two most basic visual features, contrast and luminance, across pathways to enable behaviorally relevant computations.

## A post-receptor luminance gain is utilized in both ON and OFF visual pathways, but with distinct implementations

Changing visual environments impose a common challenge onto the encoding of both ON and OFF contrasts, namely the contrasts are underestimated in sudden dim light. Our work shows that visual behaviors guided by both ON and OFF pathways approach luminance invariance and are not susceptible to underestimation of contrast in sudden dim conditions. Similarly, luminance invariance has been shown in human perception of both ON and OFF contrasts, and in neural responses in cat LGN at fast time scales (**Burkhardt et al., 1984**; **Mante et al., 2005**). This argues that the implementation of a rapid luminance gain is a common feature of all visual systems, which is relevant for any species that relies on visual information for its survival in changing visual environments. In *Drosophila*, luminance information from both L1 and L3 are required for rapid luminance gain control, but the impact of the two neurons on behavior is pathway dependent. In the OFF pathway, losing either L1 or L3 function leads to a strong deviation from luminance invariance, such that the dim light stimuli are underestimated. On the contrary, ON motion-driven behavior only underestimates dim stimuli if both L1 and L3 neuron types are not functional. Furthermore, L2 neurons, which were formerly thought to be OFF-pathway inputs, contribute contrast-sensitive information to ON behavior (**Clark et al., 2011**; **Joesch et al., 2010**; **Silies et al., 2013**). Notably, ON and OFF contrast constancy is not achieved symmetrically at every processing stage. For example, in the vertebrate retina, ON RGCs encode a mixture of luminance-invariant and absolute (i.e. luminance-dependent) contrast, whereas OFF RGCs encode predominantly absolute contrast (**Idrees and Münch, 2020**). Thus, asymmetrical implementation of contrast-corrective mechanisms can be common across visual systems, too.

## All three lamina neurons are inputs to both ON and OFF pathways

Input from the three lamina neurons is differentially utilized across ON and OFF pathways. How does this fit with the established notion that L1 is an input to the ON and L2 and L3 are inputs to OFF pathways? The luminance-varying stimuli sets used here were able to pull out lamina neuron contributions that were not obvious with simpler stimuli. For example, our data show that L1 and L2 provide redundant contrast input to the ON pathway at 100% contrast and varying luminance. However, L1 is still strictly required for ON responses if different contrasts are mixed. This is consistent with a more complex ON-pathway input architecture and hints at a role for the L1 pathway in contrast adaptation. Interestingly, Mi1, an important post-synaptic partner of L1, shows an almost instantaneous and strong contrast adaptation (**Matulis et al., 2020**).

While all three lamina neuron types hyperpolarize to light onset and depolarize to light offset, contrast selectivity emerges downstream of these neurons: post-synaptic partners of L1 acquire ON contrast selectivity due to inhibitory glutamatergic synapses, whereas cholinergic L2 and L3 synapses retain OFF contrast selectivity (**Molina-Obando et al., 2019**; **Yang et al., 2016**). L3 had furthermore mostly been considered an OFF-pathway neuron because the OFF-pathway neuron Tm9 receives its strongest input from L3 (**Fisher et al., 2015**; **Shinomiya et al., 2014**; **Takemura et al., 2013**). However, L3 itself actually makes most synaptic connections with the Mi9 neuron that plays a role in guiding behavioral responses to ON stimuli (**Strother et al., 2017**; **Takemura et al., 2013**). Further synapses of L3 with the ON-selective Mi1 neuron are similar in number to those with Tm9 (**Takemura et al., 2013**). Finally, L3 can potentially also convey information to the chromatic pathway, as Tm20 is its second strongest postsynaptic connection (**Lin et al., 2016**). There, L3 luminance sensitivity might play a relevant role in achieving color constancy, that is color recognition irrespective of illumination

conditions. Altogether, anatomical and functional data indicate that it is time to redefine L3 as part of a luminance-encoding system rather than a mere OFF-pathway input. Other synaptic connections that link L2 to downstream ON-selective neurons still have to be investigated in detail.

A role of L1 beyond the ON pathway is supported by functional connectivity studies showing that Tm9 properties rely in part on L1 input (*Fisher et al., 2015*), and that Tm9 together with other OFF-pathway interneurons displays contrast-opponent receptive fields, showing the presence of ON information in the OFF pathway (*Ramos-Traslosheros and Silies, 2021*). Connectomics data did not identify any known OFF-pathway neurons postsynaptic to L1, but among the strongest postsynaptic partners of L1 are the GABAergic interneurons C2 and C3 that connect to the OFF pathway (*Takemura et al., 2013*). Intercolumnar neurons downstream of L1, such as Dm neurons (*Nern et al., 2015*), could further carry information to OFF-selective neurons, likely through disinhibition from ON-selective inputs. In the vertebrate retina, intercolumnar amacrine cells mediate interaction between ON and OFF bipolar cells, which has been shown to extend the operating range of the OFF pathway (*Manookin et al., 2008*; *Odermatt et al., 2012*).

Altogether, it now becomes evident that a split in ON and OFF circuitry only truly exists in downstream medulla neurons and direction-selective cells. The luminance and contrast features encoded differently in L1, L2 and L3 lamina neurons are shared by both pathways. Importantly, the distinct features that are passed on by the specific inputs downstream of photoreceptors guide distinct behavioral roles.

## Neurons postsynaptic to photoreceptors encode contrast and luminance differently

Despite being postsynaptic to the same photoreceptor input, L1, L2, and L3 all show different contrast and luminance sensitivities. L1 was previously considered the ON-pathway sibling of the contrast-sensitive L2, both with regard to its temporal filtering properties and at the transcriptome level (*Clark et al., 2011*; *Tan et al., 2015*). However, L1 calcium signals show a transient and a sustained response component, which are contrast- and luminance-sensitive, respectively. Compared to photoreceptors, which also carry both contrast and luminance components, L1 still amplifies the contrast signals received from the photoreceptors, since its transient component is more pronounced than the one seen in the photoreceptor calcium traces (*Gür et al., 2020*). In other insect species, different types of lamina neurons have also been distinguished based on their physiological properties (*Rusanen et al., 2018*; *Rusanen et al., 2017*), although their specific luminance and contrast sensitivities are yet unknown.

The two luminance-sensitive neurons L1 and L3 differ in their luminance-encoding properties. L1's initial transient contrast response might reduce the operating range of the subsequent luminance-sensitive baseline. L3's calcium responses show little adaptation and can utilize most of its operating range to encode luminance. L3 seems to invest this wider operating range into amplifying the darkest luminance values selectively and non-linearly. Thus, a predominantly luminance-sensitive channel among LMCs may have evolved to selectively process stimuli in the low luminance range. The different linear and non-linear properties of L1 and L3 might further increase the dynamic range of luminance signaling (*Odermatt et al., 2012*). Together with the pure contrast sensitivity of L2, the first-order interneurons in flies exhibit a wide range of sensitivities with respect to contrast and luminance, and different functional relevance. Diversifying feature encoding through distinct temporal properties of first-order interneurons is a strategy employed to reliably handle wide luminance ranges.

## Similarities and differences of peripheral processing strategies across species

In flies, three first-order interneurons feed contrast and luminance information into downstream circuitry. In the mouse retina, more than 30 functionally distinct bipolar types show a spectrum of temporal filter properties rather than a strict transient-sustained dichotomy, thus capturing a larger diversity of temporal information in parallel channels (e.g. *Baden et al., 2016*; *Ichinose et al., 2014*; *Odermatt et al., 2012*). Many bipolar cell types resemble L1, in that they have both luminance and contrast signals in distinct response components (e.g. *Oesch and Diamond, 2011*). However, the degree of transiency varies from cell type to cell type, and some predominantly sustained bipolar cell types are also found, closely resembling the luminance-sensitive L3 (e.g. *Awatramani and Slaughter,*

*2000*; *Ichinose et al., 2014*). Such diversification of feature extraction at the periphery has been shown to be computationally advantageous, especially when processing complex natural scenes (e.g. *Odermatt et al., 2012*; *Rieke and Rudd, 2009*). For example, during daylight, visual scenes can differ in intensity by 4–5 log units, whereas electrical signals in cone photoreceptors reach a dynamic range of only two orders of magnitude (*Naka and Rushton, 1966*; *Normann and Perlman, 1979*; *Pouli et al., 2010*; *Schnapf et al., 1990*).

Although the vertebrate retina apparently has a much larger diversity of cell types to handle the wide and complex statistics of the visual environments, there is only a single layer of processing between photoreceptors and the first direction-selective cells, whereas in insects, there are two: the lamina and the medulla. It seems as if the combined properties of bipolar cells are spread across these two processing stages in the fly visual system: whereas some properties, such as diversity of temporal filtering starts in LMCs, contrast selectivity only emerges in medulla neurons and not directly in the first-order interneurons as it happens in bipolar cells. In both vertebrates and invertebrates, the emergence of ON selectivity occurs through inhibitory glutamatergic synapses, but whereas this happens at the photoreceptor-to-bipolar cell synapse in vertebrates, it happens one synapse further down between lamina and medulla neurons in flies (*Masu et al., 1995*; *Molina-Obando et al., 2019*). Taken together, LMCs and downstream medulla neurons combined appear to be the functional equivalents of vertebrate bipolar cell layers. Given the size limitations of the fly visual system to encode the same complex environment effectively, one benefit of this configuration with an extra layer could be that it allows more combinations. Furthermore, the photoreceptor-to-lamina synapse in the fly superposition eye already serves to spatially pool information from different photoreceptors (*Braitenberg, 1967*;

**Table 1.** Genotypes used in this study.

| Name | Genotype | Figure |
|---|---|---|
| **Imaging** | | |
| *L1 >>GCaMP6 f* | *w+; L1^{c202a}-Gal4 /+; UAS-GCaMP6f /+* | *Figure 1, Figure 2—figure supplement 1* |
| *L2 >>GCaMP6 f* | *w+; UAS-GCaMP6f /+; L2^{21Dhh}-Gal4 /+* | *Figure 2, Figure 2—figure supplement 1* |
| *L3 >>GCaMP6 f* | *w+; L3^{MH56}-Gal4 /+; UAS-GCaMP6f /+* | *Figure 2, Figure 2—figure supplement 1* |
| **Behavior** | | |
| UAS-shibire^{ts} control | *w+; +/+; UAS-shi^{ts} /+* | *Figures 1 and 3–6, Figure 2—figure supplement 2–1, Figure 5—figure supplement 5–1* |
| L3-Gal4 control | *w+; +/+; L3^{0595}-Gal4 /+* | *Figure 4* |
| L3 silencing | *w+; +/+; L3^{0595}-Gal4 / UAS- shi^{ts}* | *Figure 4* |
| L1-Gal4 control | *w+; L1^{c202a}-Gal4 /+; +/+* | *Figures 3 and 6, Figure 3—figure supplement 1* |
| L1 silencing | *w+; L1^{c202a}-Gal4 /+; +/UAS- shi^{ts}* | *Figures 3 and 6, Figure 3—figure supplement 1* |
| L1-Gal4, L3-Gal4 control | *w+; L1^{c202a}-Gal4 /+; L3^{0595}-Gal4/+* | *Figure 4* |
| L1, L3 silencing | *w+; L1^{c202a}-Gal4 /+; L3^{0595}-Gal4/UAS- shi^{ts}* | *Figure 4* |
| ort mutant | *w+; UAS-ort /+; ort^{1}, ninaE^{1}/Df(3 R)BSC809* | *Figures 3, 4 and 6* |
| L3 ort±control | *w+; +/+; L3^{0595}-Gal4, ort^{1}, ninaE^{1} /+* | *Figure 4* |
| L3 ort rescue | *w+; UAS-ort /+; L3^{0595}-Gal4, ort^{1}, ninaE^{1}/ Df(3 R)BSC809* | *Figure 4* |
| L1 ort±control | *w+; L1^{c202a}-Gal4/+, ort^{1}, ninaE^{1}/+* | *Figures 3 and 6* |
| L1 ort rescue | *w+; UAS-ort/+; L1^{c202a}; ort^{1},ninaE^{1}/Df(3 R) BSC809* | *Figures 3 and 6* |
| L2-Gal4 control | *w+; +/+; L2^{21Dhh}-Gal4/+* | *Figure 5, Figure 5—figure supplement 1* |
| L2 silencing | *w+; +/+; L2^{21Dhh}-Gal4/UAS- shi^{ts}* | *Figure 5, Figure 5—figure supplement 1* |
| L1-Gal4, L2-Gal4 control | *w+; L1^{c202a}-Gal4/+; L2^{21Dhh}-Gal4/+* | *Figure 5, Figure 5—figure supplement 1* |
| L1, L2 silencing | *w+; L1^{c202a}-Gal4/+; L2^{21Dhh}-Gal4/UAS- shi^{ts}* | *Figure 5, Figure 5—figure supplement 1* |

*Clandinin and Zipursky, 2002*; *Kirschfeld, 1967*). In both visual systems, diversifying distinct information across several neurons could serve as a strategy to reliably respond to contrast when luminance conditions vary.

## Materials and methods

### Experimental model

All flies were raised at 25 °C and 65% humidity on standard molasses-based fly food while being subjected to a 12:12 hr light-dark cycle. Two-photon experiments were conducted at room temperature (20 °C) and behavioral experiments at 34 °C. Female flies 2–4 days after eclosion were used for all experimental purposes. Lamina neuron driver lines used for genetic silencing and *ort* rescue experiments were $L3^{0595}$-Gal4 (*Silies et al., 2013*), $L2^{21Dhh}$-Gal4 and $L1^{c202a}$-Gal4 (*Rister et al., 2007*), and *UAS-shi[ts]*, $ort^1$,$ninaE^1$ and *Df(3 R)BSC809* were from BDSC (# 44222, 1946 and 27380). Since the $ort^1$ mutant chromosomes also carries a mutation in $ninaE^1$ (*Drosophila rhodopsin1*), we used the $ort^1$ mutation in trans to a deficiency that uncovers the *ort* but not the *ninaE* locus. *UAS-ort* was first described in *Hong et al., 2006*. For imaging experiments, GCaMP6f (BDSC #42747) was expressed using $L1^{c202a}$-Gal4, $L2^{21Dhh}$-Gal4 (*Rister et al., 2007*), and $L3^{MH56}$-Gal4 (*Timofeev et al., 2012*). Detailed genotypes are given in *Table 1*.

### Behavioral experiments

Behavioral experiments were performed as described in *Ketkar et al., 2020*. In brief, all experiments were conducted at 34 °C, a restrictive temperature for *shibire*[ts] (*Kitamoto, 2001*). Female flies were cold anesthetized and glued to the tip of a needle at their thorax using UV-hardened Norland optical adhesive. A 3D micromanipulator positioned the fly above an air-cushioned polyurethane ball (Kugel-Winnie, Bamberg, Germany), 6 mm in diameter, and located at the center of a cylindrical LED arena that spanned 192° in azimuth and 80° in elevation (*Reiser and Dickinson, 2008*). The LED panels arena (IO Rodeo, CA, USA) consisted of 570 nm LEDs and was enclosed in a dark chamber. The pixel resolution was ~2° at the fly's elevation. Rotation of the ball was sampled at 120 Hz with two wireless optical sensors (Logitech Anywhere MX 1, Lausanne, Switzerland), positioned toward the center of the ball and at 90° to each other (setup described in *Seelig et al., 2010*). Custom written C#-code was used to acquire ball movement data. MATLAB (Mathworks, MA, USA) was used to coordinate stimulus presentation and data acquisition. Data for each stimulus sequence were acquired for 15–20 min, depending on the number of distinct epochs in the sequence (see 'visual stimulation' for details).

### Visual stimulation for behavior

The stimulation panels consist of green LEDs that can show 16 different, linearly spaced intensity levels. To measure the presented luminance, candela/m² values were first measured from the position of the fly using a LS-100 luminance meter (Konika Minolta, NJ, USA). Then, these values were transformed to photons incidence per photoreceptor per second, following the procedure described by *Dubs et al., 1981*. The highest native LED luminance was approximately $11.77 * 10^5$ photons * s⁻¹ * photoreceptor⁻¹ (corresponding to a measured luminance of 51.34 cd/m²), and the luminance meter read 0 candela/ m² when all LEDs were off. For all experiments, a 0.9 neutral density filter foil (Lee filters) was placed in front of the panels, such that the highest LED level corresponded to $14.71 *10^4$ photons*s⁻¹*receptor⁻¹.

Fly behavior was measured in an open-loop paradigm where either ON or OFF edges were presented. For every set of ON or OFF edges, each epoch was presented for around 60–80 trials. Each trial consisted of an initial static pattern (i.e. the first frame of the upcoming pattern) shown for 500ms followed by 750ms of edge motion. Inter-trial intervals were 1 s. All edges from a set were randomly interleaved and presented in a mirror-symmetric fashion (moving to the right, or to the left) to account for potential biases in individual flies or introduced when positioning on the ball.

The ON edge stimuli comprised four edges, each covering 48° arena space. All ON edges moved with the angular speed of 160°/s. Thus, within a 750ms stimulus epoch, the edge motion repeated thrice: After each repetition, the now bright arena was reset to the pre-motion lower LED level, and the next repetition followed immediately, picking up from the positions where the edges terminated

in the first repetition. This way, each edge virtually moved continuously. The following sets of ON edges were presented:

1. 100% contrast edges: Here, the edges were made of 5 different luminance values (i.e. five unique epochs), moving on a complete dark background. Thus, the pre-motion LED level was zero, and the edges assumed the intensities 7%, 14%, 27%, 53%, or 100% of the highest LED intensity (corresponding to the luminances: 0.98, 1.96, 3.92, 7.84 or 14.71 $*10^4$ photons$*$s$^{-1}*$receptor$^{-1}$ luminance). Thus, every epoch comprised 100% Michelson contrast. The inter-trial interval consisted of a dark screen.
2. Single-luminance edges: 100% contrast edges of a single luminance value (0.98$*10^4$ photons$*$s$^{-1}*$receptor$^{-1}$ luminance) moved on a dark background. All epochs were identical. The inter-trial interval consisted of either a dark screen (*Figure 3—figure supplement 1A, B*) or a screen equally bright as the ON edges (*Figure 3—figure supplement 1C, D*).
3. Mixed-contrast edges: The set comprised of seven distinct epochs, each with a different Michelson contrast value (11%, 25%, 33%, 43%, 67%, 82%, and 100%). Here, the edge luminance was maintained constant at 67% of the highest LED intensity, across epochs, and the background luminance varied. The inter-trial interval showed a uniformly lit screen with luminance equivalent to the edge luminance.

For the experiments concerning OFF edges, a set of five OFF edges comprising 100% Weber contrast was used as described in *Ketkar et al., 2020*. Epoch consisted of a single OFF edge presented at one of five different uniformly lit backgrounds. The edge luminance was always ~zero, whereas the five different background luminances were 7%, 14%, 27%, 54%, and 100% of the highest LED intensity (corresponding to five different background luminances: 0.98, 1.96, 3.92, 7.84, or 14.71 $*10^4$ photons$*$s$^{-1}*$receptor$^{-1}$). The inter-trial interval consisted of a dark screen.

## Behavioral data analysis

Fly turning behavior was defined as yaw velocities that were derived as described in *Seelig et al., 2010*, leading to a positive turn when flies turned in the direction of the stimulation and to a negative turn in the opposite case. Turning elicited by the same epoch moving either to the right or to the left were aggregated to compute the mean response of the fly to that epoch. Turning responses are presented as angular velocities (rad/s) averaged across flies ± SEM. Peak velocities were calculated over the stimulus motion period (750ms), shifted by 100ms to account for a response delay, and relative to a baseline defined as the last 200ms of the preceding inter-stimulus intervals. For the moving edges of 100% contrast and varying luminance, relation between peak velocities and luminance was assessed by fitting a straight line (V = a*log(luminance) + b) to the peak velocities of individual flies and quantifying the mean slope (a) ± SEM across flies. When comparing the slopes computed for behavior and L1 physiology, the two data types were first normalized for individual flies for behavior and individual regions of interest (ROIs) for L1 physiology (*Figure 1E*). For the *ort* rescue experiments, rescue efficiency was calculated at each stimulus luminance as

$$E_{rescue} = \frac{rescue - control^-}{control^+ - control^-}$$

where $E_{rescue}$ is the fractional rescue efficiency, *rescue* is the mean peak velocity of the rescue genotype such as L1 rescue, *control*$^-$ is the mean peak velocity of the *ort* null mutant negative control and *control*$^+$ + for the mean peak velocity of the positive heterozygous *ort$^1$* control (e.g. *L1-Gal4; ort$^1$/+*). Statistical significance of $E_{rescue}$ differences was tested using a permutation test. Specifically, flies of the genotypes L1 ort rescue and L3 ort rescue were shuffled 1000 times and the difference between their rescue efficiencies was obtained each time. The difference values so obtained gave a probability distribution that approximated a normal distribution. The efficiency difference was considered significant when it corresponded to less than 5% probability on both tails of the distribution, after Bonferroni correction.

Mean turning of flies as well as the slopes from control and experimental genotypes were normal distributed as tested using a Kolmogorov-Smirnov test (p > 0.05). To test differences between these variables, pairwise t-tests considering Bonferroni-Holm correction for multiple comparisons were performed between genotypes. The experimental genotype was marked significantly different only when it differed from both genetic controls. Flies with a baseline forward walking speed of less than 2 mm/s were discarded from the analysis. This resulted in rejection of approximately 25% of all flies.

## Two-photon imaging

Female flies were anesthetized on ice before placing them onto a sheet of stainless-steel foil bearing a hole that fit the thorax and head of the flies. Flies then were head fixated using UV-sensitive glue (Bondic). The head of the fly was tilted downward, looking toward the stimulation screen and their back of the head was exposed to the microscope objective. To optically access the optic lobe, a small window was cut in the cuticle on the back of the head using sharp forceps. During imaging, the brain was perfused with a carboxygenated saline-sugar imaging solution composed of 103 mM NaCl, 3 mM KCl, 5 mM TES, 1 mM NaH2PO4, 4 mM MgCl2, 1.5 mM CaCl2, 10 mM trehalose, 10 mM glucose, 7 mM sucrose, and 26 mM NaHCO3. Dissections were done in the same solution, but lacking calcium and sugars. The pH of the saline equilibrated near 7.3 when bubbled with 95% O2 / 5% CO2. The two-photon experiments for *Figure 2* and *Figure 2—figure supplement 1* were performed using a Bruker Investigator microscope (Bruker, Madison, WI, USA), equipped with a 25 x/NA1.1 objective (Nikon, Minato, Japan). An excitation laser (Spectraphysics Insight DS+) tuned to 920 nm was used to excite GCaMP6f, applying 5–15 mW of power at the sample. For experiments in *Figure 1*, a Bruker Ultima microscope, equipped with a 20 x/NA1.0 objective (Leica, Wetzlar, Germany) was used. Here the excitation laser (YLMO-930 Menlo Systems, Martinsried, Germany) had a fixed 930 nm wavelength, and a power of 5–15 mW was applied at the sample.

In both setups, emitted light was sent through a SP680 shortpass filter, a 560 lpxr dichroic filter and a 525/70 emission filter. Data was acquired at a frame rate of ~10–15 Hz and around 6–8 x optical zoom, using PrairieView software.

## Visual stimulation for imaging

For the staircase stimuli and light flashes of different luminances, the visual stimuli were generated by custom-written software using C ++ and OpenGL and synchronized as described previously (*Freifeld et al., 2013*). The stimuli were projected onto an 8cm x 8cm rear projection screen placed anterior to the fly and covering 60° of the fly's visual system in azimuth and 60° in elevation. These experiments were performed with the Bruker Investigator microscope.

For ON-moving edges, the stimulus was generated by custom-written software using the Python package PsychoPy (*Peirce, 2008*), and then projected onto a 9cm x 9cm rear projection screen placed anterior to the fly at a 45° angle and covering 80° of the fly's visual system in azimuth and 80° in elevation. These experiments were performed with the Bruker Ultima microscope.

Both stimuli were projected using a LightCrafter (Texas Instruments, Dallas, TX, USA), updating stimuli at a frame rate of 100 Hz. Before reaching the fly eye, stimuli were filtered by a 482/18 band pass filter and a ND1.0 neutral density filter (Thorlabs). The luminance values are measured using the same procedure described above for the behavioral experiments. The maximum luminance value ($I_{max}$) measured at the fly position was $2.17*10^5$ photons $s^{-1}$ photoreceptor$^{-1}$ for the staircase and random luminance stimulation, and $2.4*10^5$ photons $s^{-1}$ photoreceptor$^{-1}$ for the ON-moving edge stimulation.

### A and B contrast steps from the adapted background

The stimulus was adapted from *Oesch and Diamond, 2011*. 30 s of bright adapting background luminance was followed by two consecutive 3 s OFF steps: the A and B steps. The A step took one of 7 decreasing luminance values, resulting in seven different contrast steps relative to the adapting step. The luminance of the B step was also composed of 7 decreasing luminance values, depending on the previous A step, resulting in 7 25% Weber contrast steps. The order of the A steps and their associated B steps was randomized.

### Staircase stimulation

The stimulus consisted of 10 s full-field flashes of five different luminances (0, 0.25, 0.5, 0.75 and 1* of the maximal luminance $I_{max}$). The different luminance epochs were presented first in an increasing order (from darkness to full brightness) then in a decreasing order (full brightness to darkness). This sequence was repeated ~3–5 times.

### Flashes of different luminances

The stimulus consisted of 10 s full-field flashes of five different luminances (0, 0.25, 0.5, 0.75 and 1* of the maximal luminance $I_{max}$). The order between the flashes was pseudo-randomized and the stimulus sequence was presented for ~300 s.

### ON moving edges at different luminances

Here, the edges were made of 6 different luminance values (corresponding to 0.16, 0.31, 0.62, 1.2, 1.8, 2.4 $*10^5$ photons$*$s$^{-1}*$receptor$^{-1}$ luminance), moving on a dark background. The inter-stimulus interval was 4 seconds of darkness.

## Two-photon data analysis

### Staircase stimulation and randomized flashes of different luminances

Data processing was performed offline using MATLAB R2019a (The MathWorks Inc, Natick, MA). To correct for motion artifacts, individual images were aligned to a reference image composed of a maximum intensity projection of the first 30 frames. The average intensity for manually selected ROIs was computed for each imaging frame and background subtracted to generate a time trace of the response. All responses and visual stimuli were interpolated at 10 Hz and trial averaged. Neural responses are shown as relative fluorescence intensity changes over time (ΔF/F0). To calculate ΔF/F0, the mean of the whole trace was used as F0. In some recordings, a minority of ROIs responded in opposite polarity (positively correlated with stimulus), as described previously (**Fisher et al., 2015**). These ROIs have their receptive fields outside the stimulation screen (**Fisher et al., 2015**; **Freifeld et al., 2013**). To discard these and other noisy ROIs, we only used ROIs that were negatively correlated (Spearman's rank correlation coefficient) with the stimulus.

To calculate the OFF-step responses to the staircase stimulus in **Figure 2B**, we first normalized the traces of L1 and L2 to get comparable values (0–1). The OFF-step response then was the difference of the maximum response and the mean of the last two seconds of the previous luminance epoch. We then fitted a sigmoidal function

$$f(x) = a * \left( \frac{1}{1+e^{k*x}} - 0.5 \right)$$

using 50 times bootstrapping with replacement to get the distribution of fit parameters (a and k).

In randomized flashes, plateau responses of neurons were calculated as the mean of the last 2 s within each luminance presentation. In the randomized flashes of different luminances, plateau response values of the highest luminance epoch were subtracted for each plateau response to get a comparable relationship between each neuron for visualization (this leads to 0 plateau response for each neuron in the highest luminance condition). Mutual information between luminance and response was calculated according to **Ross, 2014**. To characterize the distinct luminance-response relationships of L1 and L3, the difference of Pearson correlation and Spearman's rank correlation was used as a Non-linearity index. This value will reach zero if there is a strict linear relationship between luminance and response.

### A and B contrast steps from the adapted background

Data processing was performed offline using MATLAB R2019a (The MathWorks Inc, Natick, MA) following the same steps for the staircase stimulation. To calculate dF/F, the mean calcium response of the 30 s adaptation period was used as F0. Peak responses were calculated as the maximum response within each A and B steps compared to the mean baseline response in the last 2 s of the adapting period. Sustained responses were calculated as the mean response of the last 500ms of each step. We used One-way ANOVA to determine whether the peak responses to the B step were significantly different. The p-values are reported in Figure S1B.

### ON moving edges at different luminances

Data processing was performed offline using Python 2.7 (Van Rossum 1995). Motion correction was performed using the SIMA Python package's Hidden Markov Model based motion correction

algorithm (*Kaifosh et al., 2014*). The average intensity for manually selected ROIs was computed for each imaging frame and background subtracted to generate a time trace of the response. To calculate ΔF/F0, the mean of the whole trace was used as F0. The traces were then trial averaged. Responses of ROIs for each epoch was calculated as the absolute difference between the mean of the full darkness background epoch and the minimum of the ON edge presentation (minimum values are chosen because L1 neurons respond to ON stimuli with hyperpolarization).

## Statistics

Throughout the analysis procedure, mean of quantified variables were calculated first for all ROIs within a fly, and then between flies. All statistical analysis was performed between flies. For normally distributed data sets, a two-tailed Student *t* test for unpaired (independent) samples was used. For other data sets, Wilcoxon rank-sum was used for statistical analysis. Normality was tested using Lilliefors test ($p > 0.05$). One way ANOVA was used followed by multiple comparisons using the Bonferroni method for determining statistical significance between pairs of groups.

Analysis code is available at (https://github.com/silieslab/Ketkar-Gur-MolinaObando-etal2022, copy archived at swh:1:rev:392160a9f7e1336be1cb375cac53055f07620ddf; *Ketkar et al., 2022a*) and source data can be found on Zenodo: https://doi.org/105281/zenodo6335347 *Ketkar et al., 2022b*.

## Acknowledgements

We thank members of the Silies lab for comments on the manuscript. We are grateful to Christine Gündner, Simone Renner, and Jonas Chojetzki for excellent technical assistance. This project has received funding from the European Research Council (ERC) under the European Union's Horizon 2020 research and innovation program (grant agreement No 716512), from the German Research Foundation (DFG) through the Emmy-Noether program (SI 1991/1-1) and the collaborative research center 1080 "Neural homeostasis" (project C06) to MS, as well as DFG grant MA 7804/2–1 to CM.

## Additional information

### Funding

| Funder | Grant reference number | Author |
|---|---|---|
| European Commission | ERC Starting Grant No 716512 | Marion Silies |
| Deutsche Forschungsgemeinschaft | CRC1080 project C06 | Marion Silies |
| Deutsche Forschungsgemeinschaft | MA 7804/2-1 | Carlotta Martelli |

The funders had no role in study design, data collection and interpretation, or the decision to submit the work for publication.

### Author contributions

Madhura D Ketkar, Conceptualization, Investigation, Methodology, Software, Visualization, Writing – original draft, Writing – review and editing; Burak Gür, Conceptualization, Investigation, Methodology, Software, Visualization, Writing – review and editing; Sebastian Molina-Obando, Conceptualization, Investigation, Software, Visualization, Writing – original draft, Writing – review and editing; Maria Ioannidou, Investigation, Writing – review and editing; Carlotta Martelli, Methodology, Supervision, Writing – review and editing; Marion Silies, Conceptualization, Funding acquisition, Supervision, Writing – original draft, Writing – review and editing

### Author ORCIDs

Madhura D Ketkar http://orcid.org/0000-0002-0465-5616
Burak Gür http://orcid.org/0000-0001-8221-9767
Sebastian Molina-Obando http://orcid.org/0000-0003-1222-723X

Maria Ioannidou  http://orcid.org/0000-0003-2869-5468
Carlotta Martelli  http://orcid.org/0000-0002-5663-6580
Marion Silies  http://orcid.org/0000-0003-2810-9828

Decision letter and Author response
Decision letter https://doi.org/10.7554/eLife.74937.sa1
Author response https://doi.org/10.7554/eLife.74937.sa2

## Additional files

### Supplementary files
• Transparent reporting form

### Data availability
Analysis code is available at https://github.com/silieslab/Ketkar-Gur-MolinaObando-etal2022 (copy archived at swh:1:rev:392160a9f7e1336be1cb375cac53055f07620ddf), and source data can be found on Zenodo: https://doi.org/10.5281/zenodo.6335347.

The following dataset was generated:

| Author(s) | Year | Dataset title | Dataset URL | Database and Identifier |
|---|---|---|---|---|
| Ketkar, Madhura Dinesh Johannes-Gutenberg University Mainz ; Gür, Burak; Molina-Obando, Sebastian; Ioannidou, Maria; Martelli, Carlotta; Silies, Marion | 2022 | First-order visual interneurons distribute distinct contrast and luminance information across ON and OFF pathways to achieve stable behavior | https://doi.org/10.5281/zenodo.6335347 | Zenodo, 10.5281/zenodo.6335347 |

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
