## [Editor Report]

This paper combines silencing and rescue experiments with measurements of cellular responses and behavior to investigate how three early visual neurons in the fly eye encode both scene luminance and scene contrast. It reveals that these neurons carry different information about scene luminance and contrast that gets distributed to ON and OFF selective pathways that guide behavior.

---

## [Decision Letter]

**Decision letter after peer review:**

Thank you for submitting your article "First-order visual interneurons distribute distinct contrast and luminance information across ON and OFF pathways to achieve stable behavior" for consideration by *eLife*. Your article has been reviewed by 3 peer reviewers, and the evaluation has been overseen by a Reviewing Editor and Claude Desplan as the Senior Editor. The following individual involved in review of your submission has agreed to reveal their identity: Jan Ache (Reviewer #1).

Essential revisions:

The reviewers had several consensus critiques of the current manuscript, which should be addressed in revision.

1) The writing should be made clearer. As noted in the detailed reviews, there are several cases in which terminology seems unclear. Moreover, it's not always clear what is new to this paper vs. in prior work on the OFF-edge pathway or Silies et al., 2013.

2) There are several sweeping descriptions of data and conclusions about data that seem not to be entirely supported, for instance about luminance invariance. The reviewers suggest that these (which are mostly noted in the reviews) should be toned down or have nuance added so that they are commensurate with the support from the data.

3) All reviewers agreed that the luminance dependence of the behavior could be better quantified by the slope of the graphs (or other regression). This would make it more clear whether or not certain behavioral effects with silencing are real.

4) Some conflicting results need more explanation. In particular, Figure S1 shows strong attenuation of 100% contrast responses when L1 is silenced, different from the luminance sweep data in Figure 3A. Similarly, the silencing of L1 in several other papers has shown decreased turning to moving light edges, seemingly in contradiction to the data in 3A. How are these differences explained?

5) Statistics. Corrections should be made for multiple comparisons, or appropriate tests should be used.

6) One major new result in this paper compared to prior work from this group is the analysis of L1, including the claim that it encodes both luminance and contrast in distinct response components. However, the stimulus used here did not tease apart these two components, as prior published experiments had in L2 and L3. The authors should address this issue.

*Reviewer #1 (Recommendations for the authors):*

The main area of this manuscript that would benefit from improvements is the writing. The manuscript could be edited for clarity, and the language polished in several places.

*Reviewer #2 (Recommendations for the authors):*

I enjoyed this paper and I would support its publication in *eLife*.

*Reviewer #3 (Recommendations for the authors):*

The requirements/sufficiency for each LMC in the behavior, which is well documented, vs the claims about specific signals involved seems disjointed in this manuscript. In particular, the foundation of the main argument, about constancy of the behavior over luminance levels, is not quite convincing in the first place, and therefore It's hard to follow what the overall message is. Just looking at the silencing experiments on their own, one could simply say that some LMCs seems to be required at all luminance levels (L1) and others only in dim conditions (L2, L1+L3), does this imply a role of luminance responses in constancy? Some of these ideas might need to be better fleshed out.

Some specific comments:

The abstract combines finding from Ketkar et al., 2020 with those from this work. It's very confusing. In addition, I am not convinced that this manuscript shows that "responses to [both] ON [and OFF] stimuli rely on luminance based correction provided by L1 and L3".

Figure 1: The title of the figure states that L1 contrast responses don't explain ON behavior, but it is luminance that is varied between the different plots in D, and I assume the absolute peak response is plotted in E ("contrast"?). Since we later find out that L1 has 2 components (transient + sustained), this plot is confusing. Wouldn't it have made more sense to compare sustained signals in L1 to behavior? In addition, anti-correlation does not rule out causality. But in general, it's hard to understand how Gcamp responses could "explain" the behavior.

Figure 2: It would be helpful to clarify why the dF/F scales are so different between A and B. I am assuming that difference in scale is the reason we don't see decreases in dF/F to ON steps. It would be good to point this out. In addition, as mentioned in the major comments, the figure title points at differential contrast and luminance encoding but there is no analysis of contrast encoding. Doing a more thorough analysis, including contrast encoding, would go a long way towards supporting the authors' claims of distribution of signals. In particular, it would very interesting to analyze contrast encoding (transient response) as a function of "baseline" luminance (sustained response prior to the step up or down), as they don't seem to be independent.

Figure 4: It is clear from D-E that L3 rescues the behavior at lower luminances. However, the authors comment in lines 265-266 " We found that L3 is a second luminance input to the ON-pathway, that, together with L1, supports luminance-invariant responses in ON behavior". I don't see how the data supports this statement. In addition in lines 268-270: "turning responses of flies lacking both L1 and L3 functional outputs were no longer luminance invariant […] (Figure 4F,G)". All three lines in Figure 4G either increasing or decreasing with luminance increase. The absolute slope of the double mutant is the same as the control (4H). This doesn't fit with the conclusions. This confusion might stem from the definition of "luminance invariance" and the fact that from 4E and G, the controls don't seems to be luminance invariant. From G it seems like L1 and L3 are not required for behavior at higher luminance but are at lower luminance.

Figure 5: Silies et al., 2013, already show that silencing L2 decreases behavioral responses to ON edges and that L1/L2 silencing abolishes responses to ON edges. I am not sure I understand the motivation for including these data, in the context of this manuscript (in the main figures at least).

Additionally:

L1 and L3 rescues rescue ON behavior, how about L2 rescues?

L1/L3 as well as L1/L2 are silenced. How about L2/L3?

---

## [Author Response]

Essential revisions:The reviewers had several consensus critiques of the current manuscript, which should be addressed in revision.1) The writing should be made clearer. As noted in the detailed reviews, there are several cases in which terminology seems unclear. Moreover, it's not always clear what is new to this paper vs. in prior work on the OFF-edge pathway or Silies et al., 2013.

We thank all reviewers for their detailed comments on the manuscript, which helped us to improve the writing.

Here are the key new results with respect to previous papers:

– We show that ON contrast elicits behavioral responses that are near luminance invariant.

– In the OFF-pathway, luminance invariance requires a luminance signal encoded in L3. How can the ON pathway implement a luminance gain if its sole major input is the contrast sensitive L1 neuron? Connectomics data suggested that L3 feeds information into the ON pathway, and here we provide the first functional evidence for the role of L3 in ON contrast computation.

– Even though Silies et al., 2013 had suggested that L1 also provides input to the OFF pathway, here we elucidate its functional role for implementing a luminance gain. The previous suggestion from Silies et al., 2013 was built on silencing experiments, and it is more striking that individual inputs can even be sufficient in both pathways.

– Our work leads to a redefinition of the function of lamina interneurons as computational units which provide asymmetric but not exclusive contributions to ON and OFF pathways.

– Figures 1, 3, 4, and 5 address luminance invariance in the ON pathway, which does not overlap with any previous work.

2) There are several sweeping descriptions of data and conclusions about data that seem not to be entirely supported, for instance about luminance invariance. The reviewers suggest that these (which are mostly noted in the reviews) should be toned down or have nuance added so that they are commensurate with the support from the data.

We carefully edited our claims to now highlight the need for a luminance gain to scale behavioral responses to contrast. The data fully support this conclusion, even where the behavior is not entirely luminance invariant. We thank the reviewer for their detailed comments, which we individually address below.

3) All reviewers agreed that the luminance dependence of the behavior could be better quantified by the slope of the graphs (or other regression). This would make it more clear whether or not certain behavioral effects with silencing are real.

All figures showing wild type or genetic silencing data of luminance-sensitive inputs contain an analysis of the luminance dependence of the behavior, by quantifying the slopes of the graphs.

From sufficiency experiments (ort rescues) and L2 silencing experiments we only draw conclusions about contributions of individual neurons to ON and OFF pathways, and therefore did not quantify the luminance dependence in these experiments.

4) Some conflicting results need more explanation. In particular, Figure S1 shows strong attenuation of 100% contrast responses when L1 is silenced, different from the luminance sweep data in Figure 3A. Similarly, the silencing of L1 in several other papers has shown decreased turning to moving light edges, seemingly in contradiction to the data in 3A. How are these differences explained?

We were also intrigued by the discrepancy between the L1 silencing results, both across the stimulus protocols tested in this study and across studies (Clark et al., 2011; Joesch et al., 2010; Silies et al., 2013). We thus tested two further stimuli now included in Figure 3figure supplement 1 that provide an explanation. First, we presented moving ON edges of a single luminance, interleaved by dark inter-stimulus intervals and found that the changing ON edge luminance does not explain the observed discrepancy (Figure 3—figure supplement 1A,B). We next repeated this experiment with a bright inter-stimulus interval and found a remarkable response deficit in L1-silenced flies (Figure 3—figure supplement 1C,D). Notably, the previous behavioral studies as well as our mixed-contrast protocols (previously Supp. Figure 1) use bright inter-stimulus interval. Thus, the new results pinpoint a stimulus parameter that differs between the protocols leading to different phenotypes and rule out any problem with our experimental genotype. Changing the inter-stimulus intervals might alter luminance and contrast statistics of the stimulus protocol, and adaptation to these likely plays a role, however a mechanistic explanation to this is out of the scope of this manuscript. Owing to this, we removed the findings of the mixed-contrast protocol to avoid further confusion.

This section now reads:

“This was initially surprising, considering that previous behavioral studies identified L1 as the major input to the ON pathway (Clark et al., 2011; Silies et al., 2013). L1 silenced flies also turned normally to ON edges of fixed luminance, ruling out the possibility that changing luminance underlies this inconsistency (Figure 3—figure supplement 1A,B). However, L1 silencing severely reduced turning responses when a bright instead of a dark inter-stimulus interval was used, explaining the discrepancy between this and previous studies (Figure 3—figure supplement 1C,D). Thus, L1 is indeed a major but not the sole input to the ON pathway.”

5) Statistics. Corrections should be made for multiple comparisons, or appropriate tests should be used.

We now performed corrections for multiple comparisons, and explicitly state the statistical tests and methods of correction used in every figure legend as well as in the methods.

6) One major new result in this paper compared to prior work from this group is the analysis of L1, including the claim that it encodes both luminance and contrast in distinct response components. However, the stimulus used here did not tease apart these two components, as prior published experiments had in L2 and L3. The authors should address this issue.

To tease apart the contrast component more systematically, we now also include a stimulus in which we systematically change luminance while keeping contrast constant, as previously done for L2 and L3 in Ketkar et al., 2020. This is now shown as a new Figure 2figure supplement 1. To summarize, we found that L1 neuron responses contain a contrastencoding peak response and a luminance encoding sustained response. We furthermore added a comparison between the contrast-encoding step responses of L1 and L2 (Figure 2B-D), in addition to the previously shown comparison between luminance encoding by all lamina neurons (now Figure2 E-H) that revealed differences between L1 and L3 luminance encoding.

Reviewer #3 (Recommendations for the authors):The requirements/sufficiency for each LMC in the behavior, which is well documented, vs the claims about specific signals involved seems disjointed in this manuscript. In particular, the foundation of the main argument, about constancy of the behavior over luminance levels, is not quite convincing in the first place, and therefore It's hard to follow what the overall message is. Just looking at the silencing experiments on their own, one could simply say that some LMCs seems to be required at all luminance levels (L1) and others only in dim conditions (L2, L1+L3), does this imply a role of luminance responses in constancy? Some of these ideas might need to be better fleshed out.

Thank you for this feedback. As stated above (see answer to major concern 1/) we have extensively edited the manuscript according to the reviewers’ suggestions, as well as addressed the specific comments below.

We no longer assign specific behavioral roles to individual response components of neurons (e.g. contrast-sensitive peak, luminance-sensitive plateau). We find quantification of the slopes useful for describing systematic changes across luminances, instead of describing the neurons’ individual requirements at specific luminances. As highlighted above, we now describe this better in the Results section.

Some specific comments:The abstract combines finding from Ketkar et al., 2020 with those from this work. It's very confusing. In addition, I am not convinced that this manuscript shows that "responses to [both] ON [and OFF] stimuli rely on luminance based correction provided by L1 and L3".

As also written in response to reviewer #1, we now separated previous and current contributions more clearly in the abstract.

With respect to the specific sentence given by the reviewer, the luminance-based correction provided by L3 was indeed already shown in Ketkar et al., 2020. We could rewrite this part to say “responses to ON stimuli rely on luminance-based correction provided by L1.

Furthermore, our work shows that L1 also provides a luminance-based correction to the OFF pathway, in addition to the previous known contribution of L3 (Ketkar et al., 2020).” but prefer the current version because it is more concise. We are happy to change it if the reviewer feels strongly about it.

Figure 1: The title of the figure states that L1 contrast responses don't explain ON behavior, but it is luminance that is varied between the different plots in D, and I assume the absolute peak response is plotted in E ("contrast"?). Since we later find out that L1 has 2 components (transient + sustained), this plot is confusing. Wouldn't it have made more sense to compare sustained signals in L1 to behavior? In addition, anti-correlation does not rule out causality. But in general, it's hard to understand how Gcamp responses could "explain" the behavior.

It is generally thought that the peak response of these neurons encodes contrast (Laughlin et al., 1987; Laughlin and Hardie, 1978) and that (fly) behavior scales with contrast. We therefore think that the comparison shown here is valid, but we changed the title of the figure to say, “Fly behavioral responses to ON contrast do not co-vary with L1 responses.” and hope that this conveys the message more clearly. This should also avoid the reader to believe that we are “concluding” or “explaining” a behavioral role from GCaMP responses, but that calcium imaging data in Figure 1 and 2 were meant to generate hypotheses about the behavioral role of these neurons, which are subsequently tested and supported by 4 figures of data that dissect the behavioral contribution of these neurons.

Figure 2: It would be helpful to clarify why the dF/F scales are so different between A and B. I am assuming that difference in scale is the reason we don't see decreases in dF/F to ON steps. It would be good to point this out. In addition, as mentioned in the major comments, the figure title points at differential contrast and luminance encoding but there is no analysis of contrast encoding. Doing a more thorough analysis, including contrast encoding, would go a long way towards supporting the authors' claims of distribution of signals. In particular, it would very interesting to analyze contrast encoding (transient response) as a function of "baseline" luminance (sustained response prior to the step up or down), as they don't seem to be independent.

The dF/F scales mainly differ because there are bigger luminance steps in the random luminance stimulation (now Figure 2E) compared to the staircase stimulation (Figure 2A). The random nature of the stimulus allows big steps in luminance (for example, 1 to 0 of I_max_, the full range of stimulation) whereas the staircase stimulation only has 0.25*I_max_ steps. Yet, in many example traces in the random luminance stimulation, it is still possible to see the decreases in dF/F to ON steps. We have replaced previous example traces with representative example traces showing this for both (L1 and L2 in Figure 2E). Additionally, calcium imaging might be less well suited to visualize decreases in membrane potential / decreases in calcium signal.

As stated above, we now provide an analysis of the contrast encoding properties of L1 and L2 to support our claims.

Figure 4: It is clear from D-E that L3 rescues the behavior at lower luminances. However, the authors comment in lines 265-266 " We found that L3 is a second luminance input to the ON-pathway, that, together with L1, supports luminance-invariant responses in ON behavior". I don't see how the data supports this statement. In addition in lines 268-270: "turning responses of flies lacking both L1 and L3 functional outputs were no longer luminance invariant […] (Figure 4F,G)". All three lines in Figure 4G either increasing or decreasing with luminance increase. The absolute slope of the double mutant is the same as the control (4H). This doesn't fit with the conclusions. This confusion might stem from the definition of "luminance invariance" and the fact that from 4E and G, the controls don't seems to be luminance invariant. From G it seems like L1 and L3 are not required for behavior at higher luminance but are at lower luminance.

We agree with the reviewer, and edited the corresponding sections as follows:

– Lines 265-266 (now 273-275) now read: “We found that L3 is a second luminance input to the ON-pathway. To ask if L3, together with L1, provides a luminance gain to scale ON behavior, we simultaneously silenced the outputs of both L1 and L3 while measuring ON behavior across luminance”

– Lines 268-270 (now 276-278) read: “However, unlike control responses which slightly deviated from luminance invariance by showing a negative correlation with

luminance, turning responses of flies lacking both L1 and L3 functional outputs were positively correlated with luminance (Figure 4F-H)”

– As mentioned in point (2) of “Essential Revisions”, we toned down our conclusion about luminance invariance and now use the term luminance gain when appropriate, including this section.

Figure 5: Silies et al., 2013, already show that silencing L2 decreases behavioral responses to ON edges and that L1/L2 silencing abolishes responses to ON edges. I am not sure I understand the motivation for including these data, in the context of this manuscript (in the main figures at least).

It is true that in Silies et al., 2013, silencing L2 has a significant deficit in turning responses to moving ON edges. However, L2 silencing does not have the same deficit in Clark et al., 2011 (using the same stimulus, and exactly the same arena) and there is no single study that claims L2 as an input to the ON pathway, including our own previous work (e.g., Silies et al., 2013).

More importantly, all the silencing experiments for all genotypes have in principle been done before, but this work uses a different experimental paradigm, in which the same contrast is interleaved with different luminances. We feel that moving L2 silencing data to the Supplementary data would break the flow of the story and left it in the main text.

Additionally:L1 and L3 rescues rescue ON behavior, how about L2 rescues?L1/L3 as well as L1/L2 are silenced. How about L2/L3?

Our experiments were motivated by specific findings, such a distinct contrast- and luminance sensitivities, and we did not attempt to check the full matrix of all possible genotypes. We therefore did not test these two genotypes in the experimental paradigm used here.